



# Data-driven automated predictions of the avalanche danger level for dry-snow conditions in Switzerland

Cristina Pérez-Guillén[1], Frank Techel[1], Martin Hendrick[1], Michele Volpi[2], Alec van Herwijnen[1], Tasko Olevski[2], Guillaume Obozinski[2], Fernando Pérez-Cruz[2], and Jürg Schweizer[1]

[1]WSL Institute for Snow and Avalanche Research SLF, Davos, Switzerland*
[2]Swiss Data Science Center, Zurich, Switzerland

**Correspondence:** Cristina Pérez Guillén (cristina.perez@slf.ch)

**Abstract.**

Even today, the assessment of avalanche danger is by large a subjective, yet data-based decision-making process. Human experts analyze heterogeneous data volumes, diverse in scale, and conclude on the avalanche scenario based on their experience. Nowadays, modern machine learning methods and the rise in computing power in combination with physical snow cover modelling open up new possibilities for developing decision support tools for operational avalanche forecasting. Therefore, we developed a fully data-driven approach to predict the regional avalanche danger level, the key component in public avalanche forecasts, for dry-snow conditions in the Swiss Alps. Using a large data set of more than 20 years of meteorological data measured by a network of automated weather stations, which are located at the elevation of potential avalanche starting zones, and snow cover simulations driven with these input weather data, we trained two random forest (RF) classifiers. The first classifier (RF#1) was trained relying on the forecast danger levels published in the avalanche bulletin. Given the uncertainty related to a forecast danger level as a target variable, we trained a second classifier (RF#2), relying on a quality-controlled subset of danger level labels. We optimized the RF classifiers by selecting the best set of input features combining meteorological variables and features extracted from the simulated profiles. The accuracy of the danger level predictions ranged between 74 % and 76 % for RF#1, and between 72 % and 78 % for RF#2, with both models achieving better performance than previously developed methods. To assess the accuracy of the forecast, and thus the quality of our labels, we relied on nowcast assessments of avalanche danger by well-trained observers. The performance of both models was similar to the accuracy of the current experience-based Swiss avalanche forecasts (which is estimated to 76 %). The models performed consistently well throughout the Swiss Alps, thus in different climatic regions, albeit with some regional differences. A prototype model with the RF classifiers was already tested in a semi-operational setting by the Swiss avalanche warning service during the winter 2020-2021. The promising results suggest that the model may well have potential to become a valuable, supplementary decision support tool for avalanche forecasters when assessing avalanche hazard.



## 1 Introduction

Avalanche forecasting, i.e. predicting current and future snow instability in time and space (McClung, 2000), is crucial to ensure safety and mobility in avalanche-prone areas. Therefore, in many countries with snow-covered mountain regions, avalanche
warning services regularly issue forecasts to inform the public and local authorities about the avalanche hazard. Even today, these forecasts are prepared by human experts. Avalanche forecasters analyze and interpret heterogeneous data volumes diverse in scale, such as meteorological observations and model output in combination with snow cover and snow instability data, covering a wide range of data qualities, and eventually, by expert judgment, conclude on the likely avalanche scenario. Hence, operational forecasting by and large still follows the approach described by LaChapelle (1980), despite the increasing relevance
of modelling approaches (Morin et al., 2020).

The key component of public avalanche forecasts is the avalanche danger level, which is usually communicated according to a five-level, ordinal danger scale (EAWS, 2021). The danger level summarizes the avalanche conditions in a given region with regard to the snowpack stability, its frequency distribution and avalanche size (Techel et al., 2020a). Accurate danger level forecasts support recreationists and professionals in their decision-making process when mitigating avalanche risk. However,
avalanche danger cannot be measured and hence also not easily be verified – and has even been described as an art based on experience and intuition (LaChapelle, 1980; Schweizer et al., 2003). To improve quality and consistency of avalanche forecasts, various statistical models (see Dkengne Sielenou et al. (2021) for a recent review) and conceptual approaches were developed. The latter, for instance, include a proposition for a structured work-flow (Statham et al., 2018) and look-up tables (e.g. EAWS, 2017; Techel et al., 2020a), both aiding forecasters in the decision-making process of danger assessment.
A major challenge when developing as well as verifying statistical models, and avalanche forecasts in general, is the lack of a measurable target variable. Since avalanche occurrence seems a logical target variable, most of the previous approaches focused on the estimation of avalanche activity using typical machine learning methods such as classification trees (Davis et al., 1999; Hendrikx et al., 2014; Baggi and Schweizer, 2009), nearest neighbors (Purves et al., 2003), support vector machines (Pozdnoukhov et al., 2008, 2011) and random forests (Mitterer and Schweizer, 2013; Möhle et al., 2014; Dreier et al., 2016;
Dkengne Sielenou et al., 2021). To build and validate these models, a substantial amount of avalanche data is required. However, avalanche catalogues are particularly uncertain and incomplete (Schweizer et al., 2020) since they rely on visual observations that are not always possible or are delayed; the only solution is to use avalanche detection systems, but such data are still scarce and/or only locally available (e.g. Hendrikx et al., 2018; Heck et al., 2019; Mayer et al., 2020).

Apart from estimating avalanche activity, few models focused on automatically forecasting danger levels. Schweizer et al.
(1992) prepared a data set for model development that included the verified danger level for the region of Davos. Based on these data, Schweizer et al. (1994) developed a hybrid expert system to assess the danger level, integrating symbolic learning with neuronal networks, and using weather and snow cover data as input parameters for the model, which correctly classified about 70 % of the cases. A similar performance was achieved by Schweizer and Föhn (1996) using an expert system approach. Brabec and Meister (2001) trained and tested a nearest-neighbor algorithm to forecast danger levels for the entire Swiss Alps
using manually observed snow and weather data from 60 stations. They reported a low overall accuracy of 52 %, probably





due to the lack of input variables related to the snow cover stability. Combining different feature sets of simulated snow cover data and meteorological variables, Schirmer et al. (2009) compared the performance of several machine learning methods (e.g. classification trees, artificial neural networks, nearest-neighbor methods, support vector machines and hidden Markov models) to predict the danger level in the region of Davos (Switzerland). Their best classifier was a nearest-neighbor model, including

the avalanche danger level of the previous day as an additional input variable, that achieved a cross-validated accuracy of 73 %.

Despite many efforts, few of the previously developed models were operationally applied due to the lack of automatic availability, the lack of transferability to other regions, or the lack of snowpack stability input, deficiencies that all limited the usefulness in operational forecasting. Moreover, most models used daily snow and weather data, manually observed at low elevations, that do not reflect avalanche conditions in the high Alpine. Today, ample data from automated weather stations and

snow cover model outputs are available (Lehning et al., 1999). The quality and breadth of these data make it suitable to apply modern machine learning methods.

Therefore, our aim is to develop an effective data-driven approach for the prediction of the regional avalanche danger level. Even though this target variable is hard to verify and intrinsically noisy, the danger level is the most relevant component for communicating the avalanche hazard. We will focus on dry-snow conditions and develop a model that can be applied to all

snow climate regions in the Swiss Alps and should have a forecast accuracy comparable to the operational experienced-based forecast. We address avalanche forecasting as a predictive supervised classification task that involves assigning a class label corresponding to the avalanche danger level to each set of meteorological and simulated snow cover data from an automatic weather station network located in Switzerland.

## 2   Data

We rely on more than 20 years of data, collected in the context of operational avalanche forecasting in the Swiss Alps, covering measured meteorological data and snow cover simulations (Sect. 2.1), as well as the regional danger level published in the avalanche forecasts (Sect. 2.2) and local assessments of avalanche danger (Sect. 2.3). The data cover the winters from 1997-1998 to 2019-2020.

### 2.1   Meteorological measurements and snow cover simulations

In Switzerland, a dense network of automatic weather stations (AWS), located at the elevation of potential avalanche starting zones, provides real-time weather and snow data for avalanche hazard assessment. These data are used by both the Swiss national avalanche warning service for issuing the public avalanche forecast as well as by local authorities responsible for the safety of avalanche-endangered settlements and infrastructure. This network, the Intercantonal Measurement and Information System (IMIS), was set up in 1996 with an initial set of 50 operational stations in the winter of 1997-1998 (Lehning et al.,

1999). It currently consists of 182 stations (2020), of which 124 are snow stations located in level terrain at locations sheltered from the wind (Fig. 1). About 15 % of the stations are situated at elevations between 1500 and 2000 m a.s.l., 61 % between
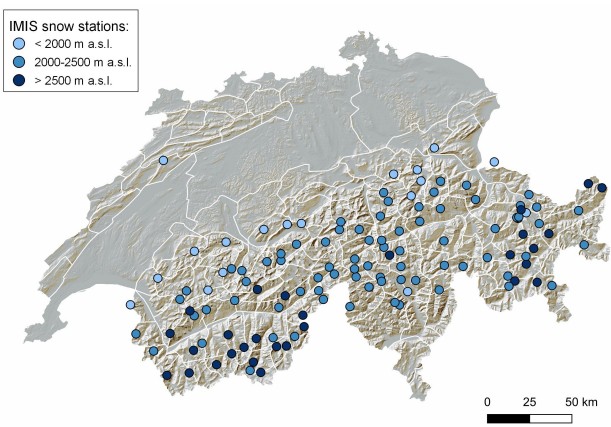

**Figure 1.** Snow stations of the IMIS network (points) located throughout the Swiss Alps (one station in northeastern Jura region), and the warning regions (white contours) used to communicate avalanche danger in the public avalanche forecast. Stations are coloured according to their elevation: below 2000 m a.s.l., between 2000 and 2500 m a.s.l. and, above 2500 m a.s.l.

2000 and 2500 m a.s.l., and 24 % between 2500 and 3000 m a.s.l. The IMIS stations operate autonomously and the data are transmitted every hour to a data server located at WSL Institute for Snow and Avalanche Research SLF (SLF) in Davos.

Based on the measurements provided by the AWS, snow cover simulations with the 1D physically-based, multi-layer model

SNOWPACK (Lehning et al., 1999, 2002) are performed automatically throughout the winter, providing output for local and regional avalanche forecasting. The meteorological data are pre-processed (MeteoIO library,  Bavay and Egger, 2014), filtering erroneous data and imputing missing data relying on temporal interpolation or on gap-filling by spatially interpolating from neighbouring stations. The SNOWPACK model provides two types of output: (1) the pre-processed meteorological data and (2) the simulated snow stratigraphy data. For an overview of the SNOWPACK model refer to Wever et al. (2014) and Morin et al.

(2020). In this study, we extracted the snow cover simulations from the database used operationally for avalanche forecasting.

## 2.2 Avalanche forecast

The avalanche forecast is published by the national avalanche warning service at SLF. During the time period analyzed, the forecast was published daily in winter - generally between early December and late April - at 1700 CET[1], valid until 1700 the following day, for the whole area of the Swiss Alps (Fig. 2). In addition, the forecast was updated daily at 0800 CET -

between about mid-December and early- to mid-April. However, only since 2013 the forecast was updated for all regions, which were assessed in the 1700 forecast (updated in the morning). Furthermore, an avalanche forecast is also published for the Jura mountains since 2017 (Fig. 2).

The Alpine forecast domain (about 26'000 km$^2$) is split into 130 warning regions (status in 2020), with an average size of about 200 km$^2$ (white polygon boundaries shown in Figs. 1 and 2). In the forecast, these warning regions are grouped

---

[1]we refer to local time, that is either CET or CEST

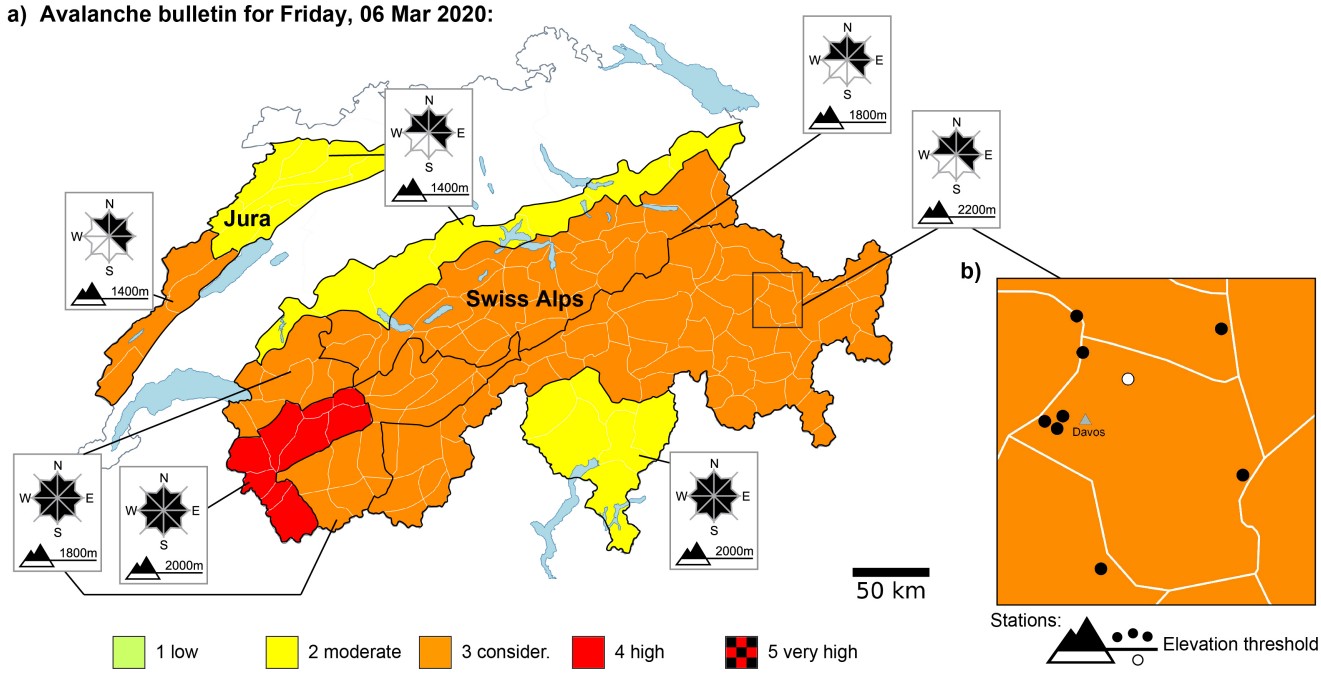

**Figure 2.** a) Map of the avalanche danger issued on Friday, 6 March 2020 at 08:00 CET. For each danger region (black contour lines), a danger level from 1-Low to 5-Very High and the critical elevations and slope aspects are graphically displayed. The white polygons show the 130 warning regions. b) Close-up map of the warning region *Davos*, with the location of the IMIS stations (points). To develop the model, we filtered days and stations as a function of the forecast critical elevation (Sect. 3.3), with stations coloured black being above this elevation on this day (here 2200 m a.s.l.), and hence considered, and the white station, located below this elevation, not being considered.

according to the expected avalanche conditions into danger regions (black polygon boundaries shown in Fig. 2). For each of these danger regions, avalanche danger is summarized by a danger level, the aspects and elevations where the danger level is valid, together with one or several avalanche problems (since 2013), and a textual description of the danger situation. The danger level is assigned according to the five-level European Avalanche Danger Scale (EAWS, 2021, levels: 1–Low, 2–Moderate, 3–Considerable, 4–High and 5–Very High).

### 2.3 Local nowcast of avalanche danger level

Specifically-trained observers assess the avalanche danger in the field and transmit their estimate to the national avalanche warning service. Observers rate the current conditions for the area of their observations, for instance after a day of backcountry touring in the mountains. To do so, they are advised to consider their own observations as well as any other relevant information (Techel and Schweizer, 2017). These assessments, called local nowcasts, are used operationally during the production of the forecast, for instance, to detect deviations between the forecast of the previous day and the actually observed conditions.





We used the local nowcasts (1) to filter potentially erroneous forecasts when compiling a subset of danger levels as described in detail in Appendix A, and (2) to discuss the model performance in light of the uncertainty related to the labels by comparing the regional forecast with the local nowcast. These assessments are human judgements and thus rely on a similar approach as a forecaster when assigning a danger level. For instance, Techel (2020) compared danger level assessments in the same area and
estimated the reliability, which is the trust we can place in an individual assessment as 0.9.

## 3 Data preparation

We first defined and prepared the target variable, the danger level (Sect. 3.1). In the next step, we extracted relevant features describing meteorological and snow cover conditions (Sect. 3.2), before labelling them with the danger levels (Sect. 3.3). Finally, we split the merged data sets for evaluating the performance of a machine learning algorithm (Sect. 3.4).

### 3.1 Preparation of target variable

We considered two approaches to define the target variable: first, by simply relying on the forecast danger level (Sect. 3.1.1), and second, by compiling a much smaller subset of «tidy» danger levels (Sect. 3.1.2). The first approach makes use of the entire database. However, this comes at the cost of potentially including a larger share of wrong labels as forecasts are inherently erroneous at times. In contrast, the second approach uses higher quality labelling, but the data size is greatly reduced.

### 3.1.1 Target variable: forecast danger level ($D_{\mathrm{forecast}}$) relating to dry-snow conditions

To train the machine learning algorithms, we rely on forecasts related to dry-snow conditions in the Alpine forecast domain (Fig. 2). Whenever a morning forecast update was available, we considered this update. In this update, on average the forecast danger level is changed in less than 3 % of the cases (Techel and Schweizer, 2017). The focus on dry-snow conditions is motivated by the fact that both the meteorological factors as well as the mechanisms that lead to an avalanche release differ greatly
between dry-snow and wet-snow avalanches. Furthermore, while danger level forecasts for dry-snow avalanche conditions are issued on a daily basis, forecasts for wet-snow avalanche conditions are only issued on days when the wet-snow avalanche danger is expected to exceed the dry-snow avalanche danger (SLF, 2020).

In total, this procedure results in forecasts issued on 3820 days during the 23 winters between 11 November 1997 and 5 May 2020, or a total of 500.545 cases (Fig. 3a). We refer to this data set as $D_{\mathrm{forecast}}$, which is used as ground truth data labeling.
The distribution of danger levels is clearly imbalanced (top of Fig. 3c). The most frequent danger levels forecast in the Alps are danger levels 2-Moderate (41 %) and 3-Considerable (36 %), which jointly account for more than 75 % of the cases. Since danger level 5-Very High is rarely forecast (<0.1 %), we merged it with danger level 4-High (2.0 %).

### 3.1.2 Compilation of subset of «tidy» danger level ($D_{\mathrm{tidy}}$)

An inherent characteristic of forecasts is that they are, at times, erroneous. The quality of avalanche forecasts is difficult to
assess, as avalanche danger is not measurable, and hence remains an expert estimate even in hindsight (Föhn and Schweizer,


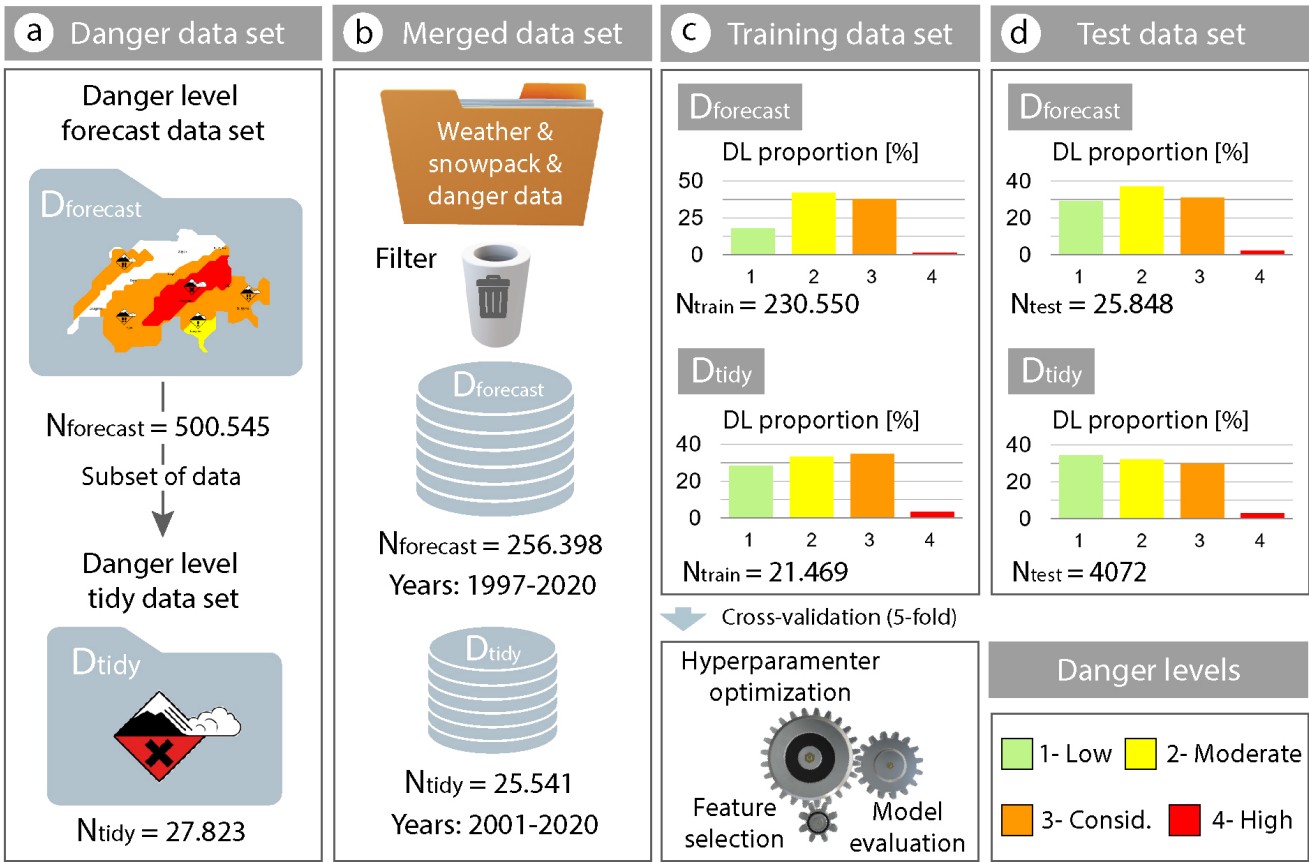

**Figure 3.** Flowchart of the data set distributions and steps, including the raw data size, the merged and filtered data set size and the danger level distributions of the training and test sets. Two machine learning classifiers are trained using as labels i) the forecast danger levels ($D_{\mathrm{forecast}}$) in the public bulletin and ii) a subset of «tidy» danger levels ($D_{\mathrm{tidy}}$). An iterative process of hyperparameter tuning and feature selection using 5-fold cross-validation was conducted to select the best model.

1995; Schweizer et al., 2003). In the case of forecast danger levels in Switzerland, recent studies comparing nowcast assessments of avalanche danger with forecast danger levels have estimated the accuracy of the forecast danger level between about 75 % and 90 %, with decreasing accuracy with increasing danger level (e.g. Techel and Schweizer, 2017; Techel, 2020). A particularly low accuracy ($<70$ %) was noted for forecasts issuing danger level 4-High (Techel, 2020). Furthermore, a strong

tendency towards over-forecasting (one level) has been noted, with forecasts rarely being lower compared to nowcast assessments of avalanche danger (e.g. Techel et al., 2020b).

Even though it is conceptually difficult to verify avalanche forecasts, and specifically a forecast danger level, we compiled a subset of re-analysed danger levels, for which we were more certain that the issued danger level was correct. This should not be considered as a verified danger level, but simply as a subset of danger levels, which presumably have a greater correspondence





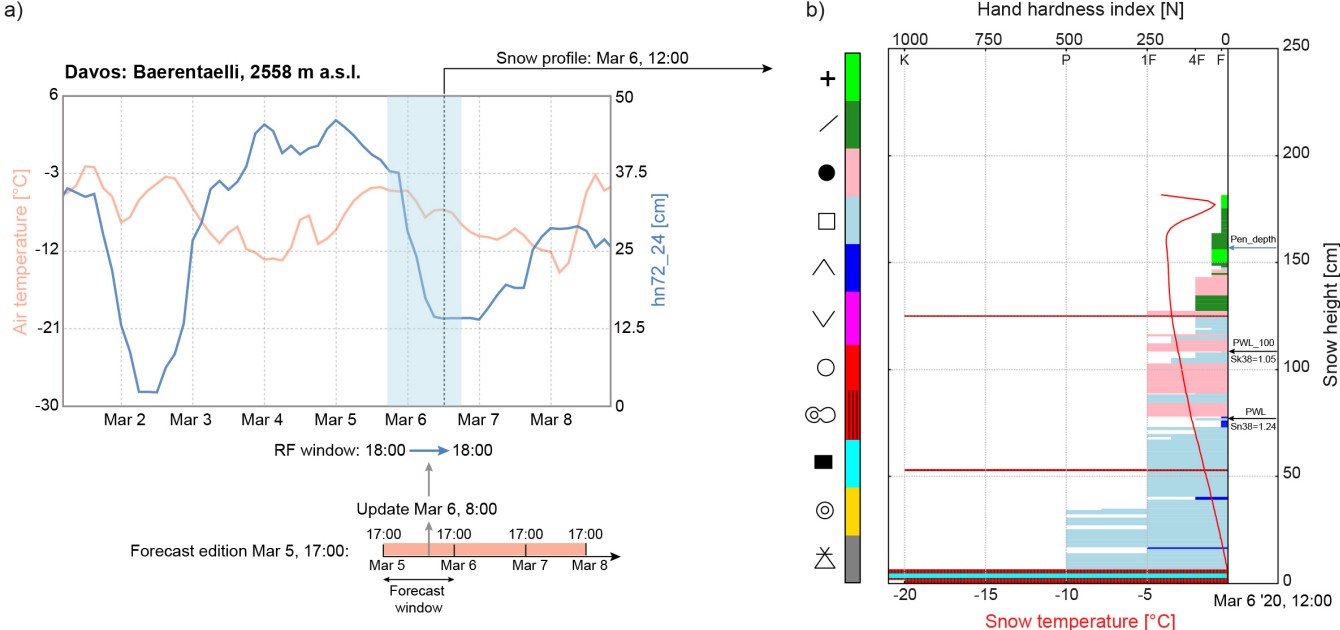

**Figure 4.** (a) Seven-day time series (March, 2020) of two meteorological features: air temperature (measured) and 3-day sum of new snow height (hn72_24, simulated by SNOWPACK) at the IMIS snow station Baerentaelli, which is located near Davos at 2558 m a.s.l. The blue area delimits an example of a 24 hours time-window (random forest, RF window) from 5 March 2020 at 18:00 to 6 March 2020 at 18:00, which is used to extract the averaged values used as inputs for the random forests algorithm. The avalanche forecast updated on 6 March 2020 at 8:00 is used for labelling the danger rating over the entire RF window. (b) Simulated snow stratigraphy from SNOWPACK at the same station on 6 March 2020 at 12:00 showing hand hardness, snow temperature and grain type (colors). Hand hardness F corresponds to fist, 4F to four fingers, 1F to one finger, P to pencil, and K to knife. Labels of grain types and colors are coded following the international snow classification (Fierz et al., 2009). The black arrows indicate the two critical weak layers located in the first 100 cm of the snow surface (PWL_100) and in a deeper layer (PWL), which were detected with the TSA approach. The blue arrows indicate the skier penetration depth (Pen_depth).

with actual avalanche conditions compared to simply using the forecast danger level. To compile this subset, we checked the forecast danger level $D_{\text{forecast}}$ by considering additional pieces of evidence. For this, we relied on

- observational data - as for instance, danger level assessments (local assessments) provided by experienced observers after a day in the field (Sect. 2.3; Techel and Schweizer, 2017) or avalanche observations, and

- the outcome from several verification studies (Schweizer et al., 2003; Schweizer, 2007; Bründl et al., 2019; Zweifel
et al., 2019).

Thus, this data set is essentially a subset of $D_{\text{forecast}}$, containing cases of $D_{\text{forecast}}$, which were either confirmed or validated following multiple pieces of evidence. Comparably few of these cases (5 %) were actually cases when the forecast danger level was corrected. These changes affected primarily days and regions when the forecast was either 4-High or 5-Very High, or the





verified danger level was one of these two levels. We refer to this subset as «tidy» danger levels ($D_{\text{tidy}}$), which is also used as
ground truth data labeling. A detailed description regarding the compilation of this data set is found in Appendix A.

$D_{\text{tidy}}$ ($N = 25.541$ cases in Fig. 3b) comprises about 10 % of the $D_{\text{forecast}}$ data set ($N = 256.398$ cases after filtering in Fig. 3b). In this subset, the distribution of the lower three danger levels is approximately balanced (about 30 % each, Fig. 3). Still, this subset contains comparably few cases of higher danger levels (4-High: 4.1 %, 5-Very High: (0.3 %). These two danger levels were again merged and labelled 4-High.

## 3.2 Feature engineering

The SNOWPACK simulations provide two different output files for each station: i) time series of meteorological variables and ii) simulated snow cover profiles. The first includes a combination of measurements (i.e. air temperature, relative humidity, snow height, or snow temperature) and of derived parameters (i.e. height of new snow, outgoing and incoming long-wave radiation, and snow drift by wind). The snow profiles contain the simulated snow stratigraphy describing layers and their properties. Fig. 4 shows an example of these data. A list of the 67 available weather and profile features is shown in Tables B1 and B2 (Appendix B).

**Meteorological input features**

The meteorological time series with 3-hour resolution are resampled to non-overlapping 24-hour averages, for a time window from 1800 of a given day to the following day at 1800 CET (24-hour window in Fig. 4a), which is the nearest to the publication time of the forecast (1700 CET).

Besides the 24-hour mean, we also trained models considering as input values the standard deviation, maximum, minimum, range and differences between subsequent 24-hour windows during the exploratory phase. However, we noted that using these additional features did not improve the overall accuracy. In addition to the data describing the day of interest, we also extracted values for the last three and seven days (Table B1). If there were missing values in the pre-processed time series, we removed these samples.

**Profile input features**

The simulated snow profiles provide highly detailed information on snow stratigraphy, as each layer is described by many parameters, and each profile may consist of dozens of layers. To reduce the complexity of the snow profile output and to obtain potentially relevant features, we extracted parameters defined and used in previous studies from the profiles at 1200 CET, which we consider the time representative of the forecast window (Fig. 4b, Table B2). These parameters included the skier penetration depth (pen_depth, Jamieson and Johnston, 1998) and snow instability variables such as the critical cut length (ccl, Gaume et al., 2017; Richter et al., 2019), the natural stability index (Sn38, Föhn, 1987; Jamieson and Johnston, 1998; Monti et al., 2016), the skier stability index (Sk38, Föhn, 1987; Jamieson and Johnston, 1998; Monti et al., 2016) and the structural stability index (SSI, Schweizer et al., 2006). We extracted the minimum of the critical cut length considering all layers below the

penetration depth (min_ccl_pen). We retrieved the instability metrics for two depths where potentially relevant persistent weak layers existed following the threshold sum approach adapted for SNOWPACK (Schweizer and Jamieson, 2007; Monti et al., 2014). We located the persistent weak layer closest to the snow surface, but within the uppermost 100 cm of the snowpack (PWL_100 in Figure 4b), and then searched the next one below (PWL in Fig. 4b). For these two layers, we extracted the parameters related to instability (ccl, Sn38, Sk38, SSI). If no persistent weak layers were found following this approach, and

to avoid missing values in the data, we assigned the respective maximum value of ccl, Sn38, Sk38, SSI observed within the entire data set, indicating the absence of a weak layer.

### 3.3 Assigning labels to extracted features

We assigned a label (danger level) to the extracted features by linking the data of the respective station with the forecast for this warning region and RF window (Fig.s 2b and 3b). Thus, each set of features extracted for an individual IMIS station (Fig.

1) was labelled with the forecast danger level for the day of interest.

Since avalanche danger depends on slope aspect and elevation, the slope aspects and elevations where the danger level prevails are described in the public forecast (Fig. 2). Outside the indicated elevation band and aspects, the danger is lower, typically one danger level (SLF, 2020). Therefore, we discarded the data from stations on days when the elevation indicated in the forecast was above the elevation of the station. If no elevation was indicated, which is normally the case at 1-Low, we

included all stations.

To further enhance the data quality, we removed data of unlikely avalanche situations. Those included data when the danger level was for 4-High, but the 3-day sum of new snow (HN72_24, Table B1) was less than 30 cm, or when the snow depth was less than 30 cm.

### 3.4 Splitting the data set

We split our data set into training and test sets corresponding to different winter seasons to ensure that training and test data were temporally uncorrelated. We defined the test set as the two most recent winter seasons of 2018-2019 and 2019-2020 (Fig. 3d). The training set corresponded to the remaining data, including the seasons from 1997-1998 to 2017-2018 (21 winters). The size of the test set is 10 % of the total amount of data and will be used for a final, unbiased evaluation of the model's generalization. We optimized the model's hyperparameters and selected the best subset of features using 5-fold cross-validation on the training

set. Each subset contains data of four/five consecutive winter seasons with an approximate size of 20 % of the training data set. This again ensures that feature selection was not affected by temporally correlated data.

### 4 Model optimization

Danger level prediction can be formulated as a supervised classification problem: given the measurements and simulations, we aim to predict the danger level on a daily basis for each station.





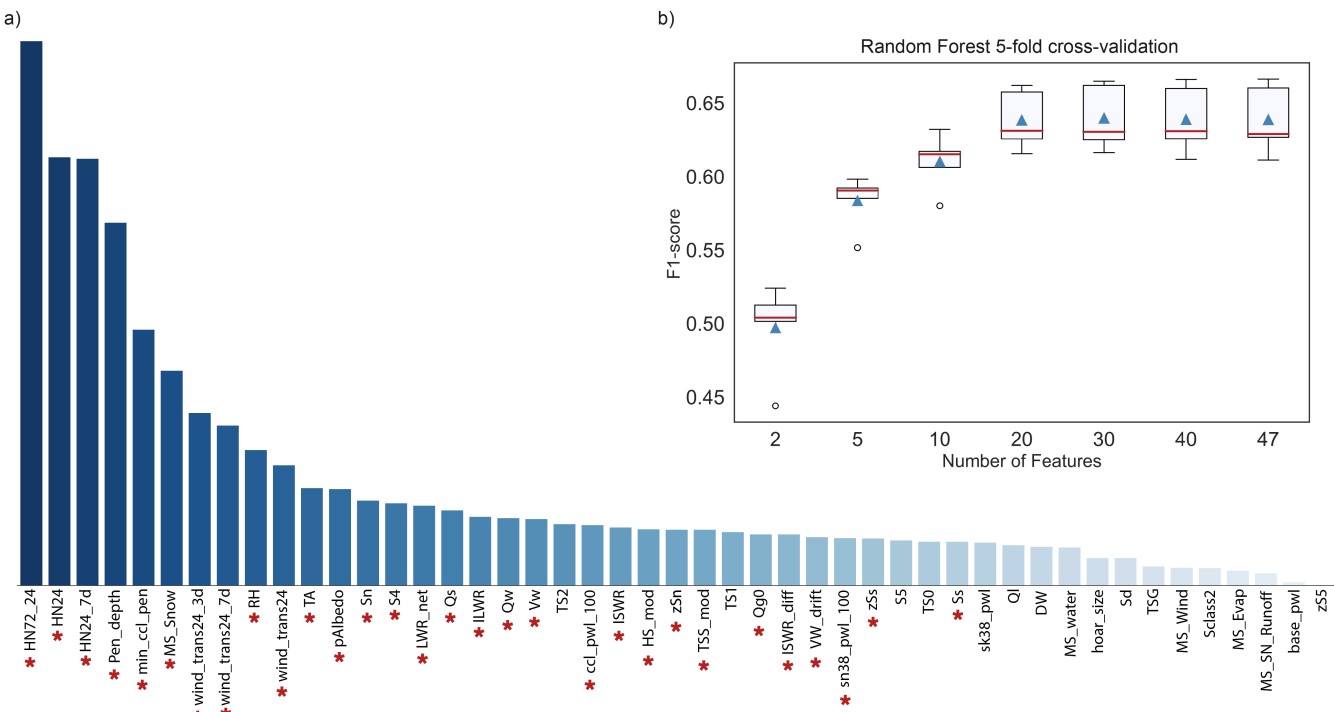

**Figure 5.** a) Feature importance ranking scored by random forest classifier (y-axis normalised). A description of each feature is shown in Tables B1 and B2 of Appendix B. The red asterisk denotes the final set of features selected to train the model. b) Box plot of the distribution of the Macro-F1 score (5-fold cross-validation) for random forest classifier with varying number of features from 2 to 47.

We tested a variety of widely used supervised learning algorithms, and the best scores were obtained with random forests (Breiman, 2001), which is among the state-of-the-art techniques for classification. Random forests are powerful, nonlinear classifiers combining an ensemble of weaker classifiers, in the form of decision trees. Each tree is grown on a different bootstrap sample containing randomly drawn instances with replacement from the training data. Besides bagging, random forests also employ random feature selection at each node of the decision tree. Each tree predicts a class membership, which can be

transformed into a probability-like score by computing the frequency at which a given test data point is classified across all the trees. The final prediction is obtained by taking a majority vote of the predictions from all the trees in the forest, or equivalently, by taking the class maximizing the probability.

     Our classification problem is extremely imbalanced; danger level 4-High (Fig. 3) accounts only for a small fraction of the whole data set. As this is representative of the actual danger level forecasts in Switzerland, we decided to build the predic-

tive model on the imbalanced problem rather than artificially sub-sampling majority classes. To achieve this, we employed a weighted impurity score to split the nodes of the trees, where the weight corresponds to the inverse of the class frequency. This ensures that prevalent classes do not dominate each split and rare classes also count towards the impurity score. We used the standard random forests implementation from the scikit-learn library (Pedregosa et al., 2011).





We trained two random forest models: RF#1 was trained using the labels from the complete data set of forecast danger

levels ($D_{\text{forecast}}$, Fig. 3b), while RF#2 was trained with the much smaller data set of «tidy» danger levels ($D_{\text{tidy}}$, Fig. 3b). We compared their performance on a common test set.

## 4.1 Model selection

We selected the best random forest model by a three-step cross-validation strategy. For this, we used cross-validation maximizing the Macro-F1 score, which corresponds to the unweighted mean of F1 scores computed for each class (danger level),

independently. F1 score is a popular metric for classification, as it balances Precision and Recall into their harmonic mean, ranging from 0 (worst) to 1 (best). Macro-F1 showed the best performance for both minority and majority classes. All the metrics used to evaluate the performance of the models are defined in Appendix B.

In the first step, we selected a set of hyperparameters from a randomized search, which maximizes the Macro-F1 score. After choosing the first optimum set of hyperparameters, we selected the best 30 input features, by ranking them according to the

feature importance score given by the random forests algorithm and the strategy described below. In the third step, we refined the hyperparameters by a dense grid search centered around the best parameters from the first step, but using the optimum feature set. This strategy shows optimal accuracy for all the classes, while keeping the model as small as possible in terms of features.

## 4.2 Feature selection

We used different approaches to remove unnecessary features and select a subset that provides high model accuracy while reducing the complexity of the model. First, variables that are strongly correlated were dropped ($\|r^2\| \geq 0.9$). For a given pair of highly correlated weather features, we removed the one showing a lower random forest feature importance score (obtained from the first step described above), which is shown in Fig. 5a. In the case of correlation between profile features, we kept the variables extracted from the uppermost weak layer that is usually more prone to triggering. A total of 20 highly correlated

variables were removed from the initial data set, leaving 47 features (Tables B1 and B2). The overall performance of the model remained the same after removing these features. In addition, we manually discarded the snow temperatures (TS0, TS1 and TS2) measured at 25 cm, 50 cm and 100 cm above ground, Fig. 5a and Table B1), as their incorporation in the model requires a larger minimum snow depth ($> 100$ cm) for meaningful measurements.

Fig. 5a shows that the features with highest importance were various sums of new snow and drifted snow, the snowfall rate,

the skier penetration depth, the minimum critical cut length in a layer below the penetration depth, the relative humidity, the air temperature and two stability indices. Hence, the highest-ranked features selected by the random forest classifier were in line with key contributing factors used for avalanche danger assessment.

To select the best subset of features, we applied the approach of Recursive Feature Elimination (RFE) (Guyon et al., 2002), which is an efficient method to select features by recursively considering smaller sets of them. An important hyperparameter

for the RFE algorithm is the number of features to select. To this end, we wrapped a random forest classifier, which was trained with a growing number of features. Features were added in descending order from the most to the least important in the score





ranking estimated by the random forest (Fig. 5a). Fig. 5b shows the variation of the mean of the Macro-F1 score with the number of selected features. The performance improves as the number of features increases until the curve levels off for 20 or more features. We selected a subset of 30 features (highest Macro-F1 score). The final set of features selected applying RFE are highlighted with a red asterisk in Fig. 5a, which are used to train the two final models RF#1 and RF#2 (complete and «tidy» data sets). Note that this last step, although it might seem redundant of the internal feature ranking made by the random forest algorithm, ensures that the growing subset of features provides consistent improvements, and the feature selection is not biased by the way the impurity score is computed (Strobl et al., 2007).

## 5 Model evaluation

In the following, we first present key characteristics describing the performance of the RF classifiers (Sect. 5.1), before exploring the average prediction accuracy on a daily basis (Sect. 5.2). To this end, we additionally consider the uncertainty related to the forecast danger level. In Sects. 5.3 and 5.4, we analyze the performance of the models in different climate regions and the impact of filtering data using the elevation information in the forecast. Finally, we assess the performance when the danger changes or stays the same (Sect. 5.5), and in case the danger level of the previous day is added as additional input feature (Sect. 5.6).

### 5.1 Performance of random forest classifiers

We developed two models: the first model, RF#1, was trained with $D_{\mathrm{forecast}}$ as labels, the second model, RF#2, was trained with $D_{\mathrm{tidy}}$. The two last winter seasons of 2018-2019 and 2019-2020 (Fig. 3d) were used as test set to calculate overall and per-class scores of these models.

When evaluating the performance against $D_{\mathrm{forecast}}$ test set, RF#1 achieved an overall accuracy (number of correctly classified samples over the total number of samples) of 0.74 and a Macro-F1 score of 0.7 (Table 1a). Even though RF#2 was trained with only 9 % of the data (Fig. 3c), it reached an almost similar overall accuracy of 0.72 and a Macro-F1 of 0.68 (Table 1b). F1 scores for each class were also fairly equal for both models (Table 1a and b). However, for the minority classes of danger levels 1-Low and 4-High, the precision of RF#1 was higher, whereas a higher proportion of samples were correctly classified by RF#2 (higher recall). This result highlights the impact of using better-balanced training data in RF#2 and less noisy labels.

The performance of the models tested on $D_{\mathrm{tidy}}$ showed that RF#2 achieved the highest Macro-F1 score of 0.75 and overall accuracy of 0.78 (Table 1d), with very similar values for RF#1 (accuracy 0.76, Macro-F1 0.74). The class breakdown for the two models showed better scores when tested against $D_{\mathrm{tidy}}$ compared to $D_{\mathrm{forecast}}$. The performance increased most notably for danger level 4-High, with the F1 score reaching 0.64.

The confusion matrices shown in Fig. 6 provide more insight into the performance of both models. The values in the diagonal clearly dominate. This indicates that the majority of cases was correctly predicted by the classifiers, as is also shown in Table 1 (the percentages shown in the diagonal correspond to the recall in Table 1). Furthermore, if predictions deviated from the ground truth label, the difference was in most cases one danger level, and only rarely two danger levels (< 3 %).





**Table 1.** Test set model performance scores of the two final random forest models (RF#1 and RF#2): precision (Prec.), recall (Rec.) and F1 for each danger level, overall accuracy (Acc.) and Macro-F1 score. a) Predictions RF#1 vs. $D_{\text{forecast}}$ (ground truth). b) Predictions RF#2 vs. $D_{\text{forecast}}$ (ground truth). c) Predictions RF#1 vs. $D_{\text{tidy}}$ (ground truth). d) Predictions RF#2 vs. $D_{\text{tidy}}$ (ground truth).

| Model:Ground truth | DL | Prec. | Rec. | F1 | Support | Model:Ground truth | DL | Prec. | Rec. | F1 | Support |
|---|---|---|---|---|---|---|---|---|---|---|---|
| | 1-Low | 0.84 | 0.79 | 0.81 | 7574 | | 1-Low | 0.73 | 0.89 | 0.81 | 7574 |
| | 2-Mod. | 0.68 | 0.66 | 0.67 | 9657 | | 2-Mod. | 0.71 | 0.57 | 0.63 | 9657 |
| | 3-Cons. | 0.72 | 0.79 | 0.75 | 8020 | | 3-Cons. | 0.73 | 0.74 | 0.74 | 8020 |
| a) RF#1: $D_{\text{forecast}}$ | 4-High | 0.63 | 0.51 | 0.57 | 597 | b) RF#2: $D_{\text{forecast}}$ | 4-High | 0.51 | 0.54 | 0.53 | 597 |
| | Acc. | | | 0.74 | 25.848 | | Acc. | | | 0.72 | 25.848 |
| | Macro-F1 | | | 0.70 | 25.848 | | Macro-F1 | | | 0.68 | 25.848 |
| | 1-Low | 0.93 | 0.78 | 0.85 | 1400 | | 1-Low | 0.87 | 0.90 | 0.88 | 1400 |
| | 2-Mod. | 0.67 | 0.70 | 0.68 | 1316 | | 2-Mod. | 0.73 | 0.67 | 0.70 | 1316 |
| | 3-Cons. | 0.73 | 0.84 | 0.78 | 1223 | | 3-Cons. | 0.76 | 0.78 | 0.77 | 1223 |
| c) RF#1: $D_{\text{tidy}}$ | 4-High | 0.64 | 0.65 | 0.64 | 133 | d) RF#2: $D_{\text{tidy}}$ | 4-High | 0.56 | 0.71 | 0.63 | 133 |
| | Acc. | | | 0.76 | 4072 | | Acc. | | | 0.78 | 4072 |
| | Macro-F1 | | | 0.74 | 4072 | | Macro-F1 | | | 0.75 | 4072 |

To analyze the model bias in more detail, we defined a model bias difference $\Delta_{DL}$ as:

$$\Delta_{\text{DL}} = DL_{\text{RF}} - DL_{\text{True}} \qquad (1)$$

where $DL_{\text{RF}}$ is the danger level predicted by the random forest model and $DL_{\text{True}}$ is the ground truth danger level. Table 2 summarizes the percentages of test samples for each model bias difference.

Compared to $D_{\text{forecast}}$, RF#1 exhibited a bias towards higher danger levels ($\sim 15\%$) rather than lower ones ($\sim 11\%$; Table 2a), while RF#2 showed an inverse trend of deviations (Table 2b). Compared with $D_{\text{tidy}}$, RF#1 showed an even larger bias towards higher danger levels (Table 2b), compared to RF#2, which had an almost equal proportion of predictions which were higher (12%) or lower (10%). Regardless which of the two model was evaluated, predictions tended to be higher at 2-Moderate ($\Delta_{DL} = 1$; between 20% and 24% in Fig. 6) and lower for 3-Considerable ($\Delta_{DL} = -1$; between 12% and 19% in Fig. 6). As 1-Low and 4-High are at the respective lower and upper end of the scale, wrong predictions can only be too high at 1-Low and too low at 4-High.

In summary, and as can be expected, each model performed better when compared to its respective test set. RF#1 achieved better performance compared to RF#2 when evaluating them on the $D_{\text{forecast}}$ test set; while RF#2 achieved slightly higher performance on $D_{\text{tidy}}$ test set. The performance of both models improved when tested against the best-possible test data ($D_{\text{tidy}}$), particularly particularly for the danger levels 1-Low and 4-High.


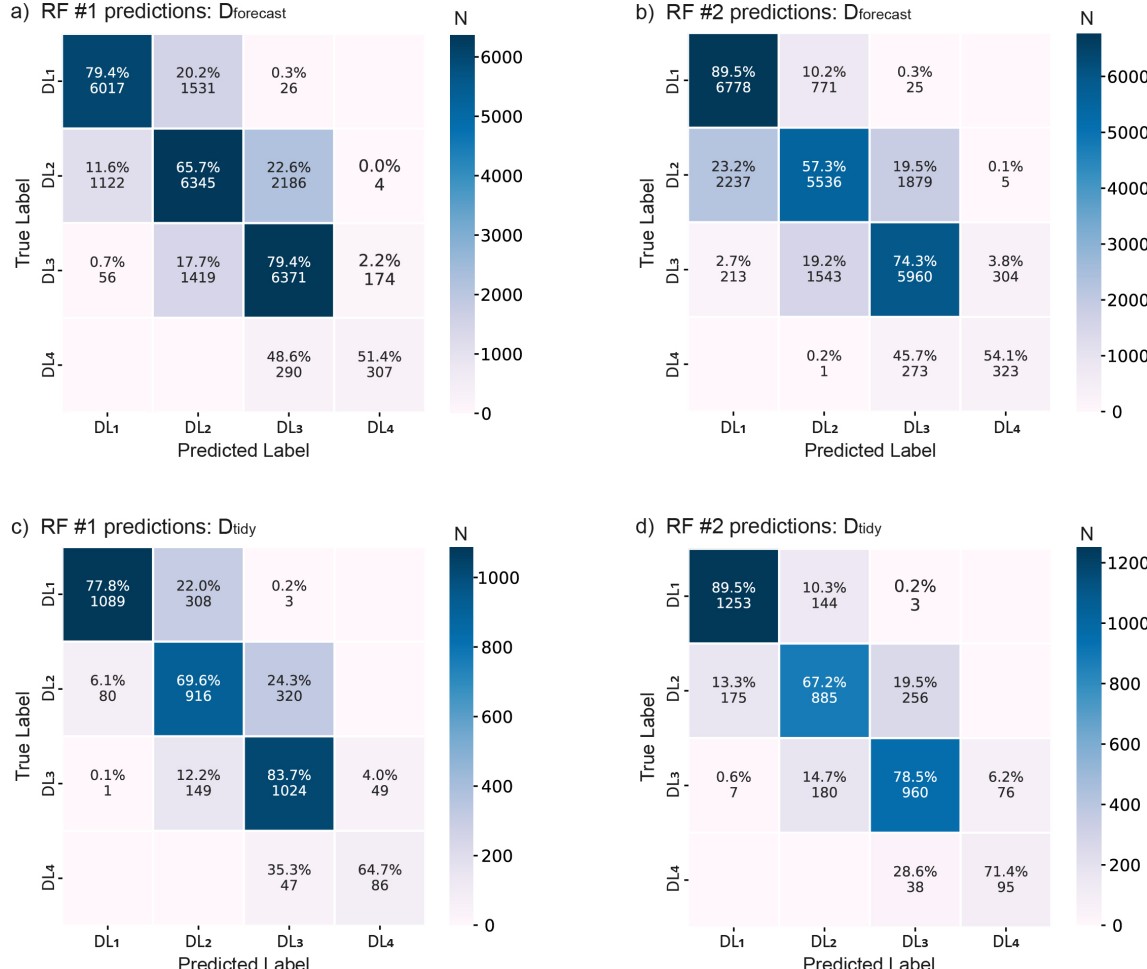

**Figure 6.** Confusion matrices of the two random forest models, RF#1 (trained with $D_{\text{forecast}}$) and RF#2 (trained with $D_{\text{tidy}}$) on the test set data of a) the forecasted danger levels and b) the «tidy» danger levels of the winter seasons of 2018-2019 and 2019-2020.

## 5.2 Daily variations in model performance and the impact of the ground truth quality on performance values

In the next step, we compare the predictive performance of the two random forest models during the two test seasons by analyzing the performance on a daily basis. To this end, we only consider the predictions using the forecast danger level ($D_{\text{forecast}}$) as the number of predictions per day is much larger than in the «tidy» data set. Nevertheless, when discussing the performance of the models, we must also consider the uncertainty related to this target variable, as errors in the ground truth can significantly impact the performance of the models. This is particularly important in our case, as we rely on the forecast 325 danger level ($D_{\text{forecast}}$) as the ground truth label. To highlight this challenging aspect of model performance evaluation, we





**Table 2.** Model (M) used for training and ground truth labels (GT) of the test set, bias ($\Delta_{DL}$) and the proportion of samples (P) for each bias value. Both models are evaluated on the $D_\text{forecast}$ test set (upper part) and $D_\text{tidy}$ test set (lower part).

| M : GT | $\Delta_{DL}$ | P [%] | M : GT | $\Delta_{DL}$ | P [%] |
|---|---|---|---|---|---|
| a) RF #1: $D_\text{forecast}$ | +2 | 0.1 | b) RF #2: $D_\text{forecast}$ | +2 | 0.1 |
| | +1 | 15.1 | | +1 | 11.4 |
| | 0 | 73.7 | | 0 | 72.0 |
| | -1 | 11.0 | | -1 | 15.7 |
| | -2 | 0.2 | | -2 | 0.8 |
| c) RF #1: $D_\text{tidy}$ | +2 | 0.1 | d) RF #2: $D_\text{tidy}$ | +2 | 0.1 |
| | +1 | 16.6 | | +1 | 11.7 |
| | 0 | 76.5 | | 0 | 78.4 |
| | -1 | 6.8 | | -1 | 9.7 |
| | -2 | 0.0 | | -2 | 0.2 |

compare the daily accuracy of the models with the «accuracy» of the forecast, which we estimate by comparing the regional forecast to the local nowcast provided by experienced observers.

The daily averaged accuracy of the predictions of the two models, the overall match between the model outputs and $D_\text{forecast}$ as ground truth, is shown in Fig. 7a. Variations in the daily accuracy of the two models were highly correlated (Pearson correlation coefficient: 0.88). The average difference in the daily accuracy between the two RF models is 0.07; on 75 % of the days it was less than 0.1. Overall, the performance of RF #1 was slightly better than RF #2 as is reflected in the overall scores (Table 1a and b), and as it can be expected when comparing with $D_\text{forecast}$ because RF #1 was trained with this data set. The match between predictions and $D_\text{forecast}$ is comparably high on about half of the days (RF #1 accuracy > 0.74, RF #2 accuracy > 0.70) and less than 0.5 on 11 % (RF #1) and 15 % (RF #2) of the days, respectively.

To estimate the accuracy of the forecast, we rely on the local nowcast reported by observers (Sect. 2.3). Thus, we consider the agreement rate between forecast danger level ($DL_F$) and nowcast danger level ($DL_\text{N}$) as a proxy for the accuracy of the forecast (e.g. Jamieson et al., 2008; Techel and Schweizer, 2017). The agreement rate ($P_\text{agree}$) for a given day is then the normalised ratio between the number of cases where nowcast and forecast agree ($N(DL_\text{F} - DL_\text{N} = 0)$) to the number of all forecast-nowcast pairs ($N$):

$$P_\text{agree} = \frac{N(DL_\text{F} - DL_\text{N} = 0)}{N} \tag{2}$$

On average, regional forecasts and local nowcasts agreed 75 % of the time ($N = 5'099$). However, considerable variations in the daily agreement rate can be noted in Fig. 7a, where the agreement rate is represented by the blue shaded area and where the points show the number of observers that provided an assessment. Considering the 171 days with more than 15 assessments,



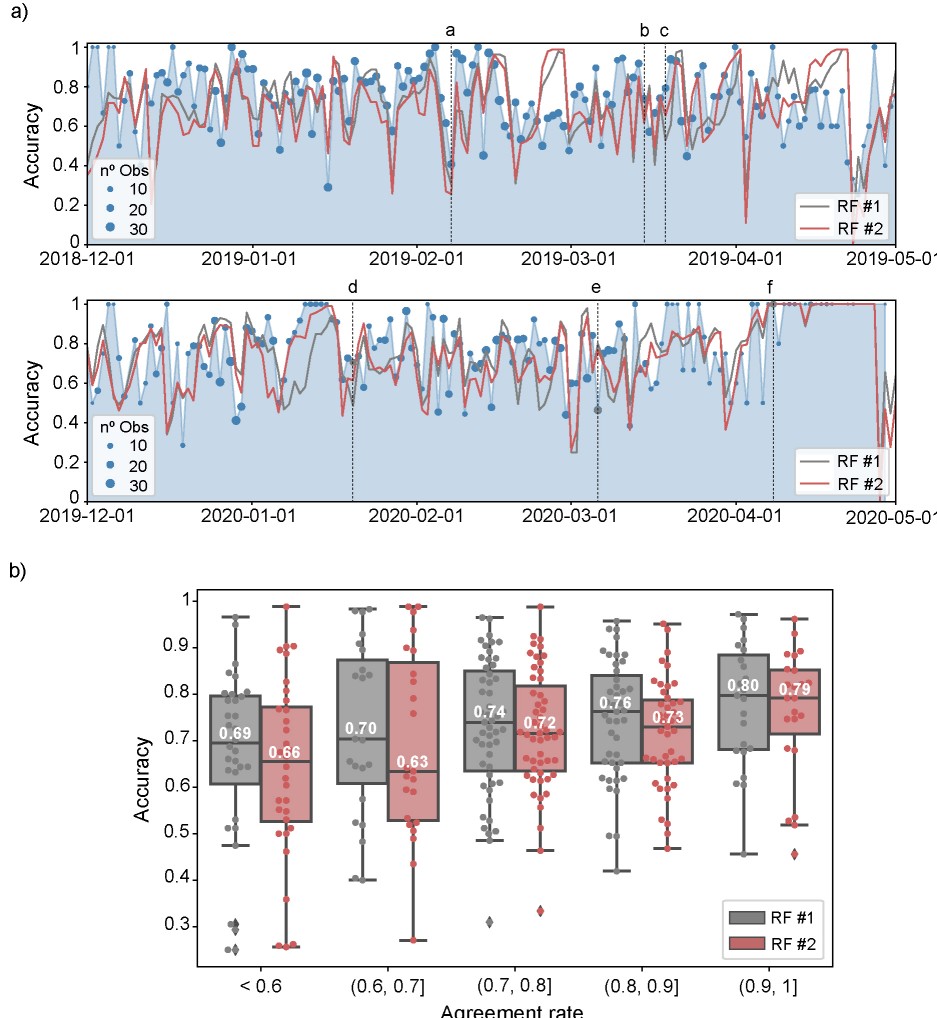

**Figure 7.** a) Comparison of the time-series of the daily averaged accuracy of the two random forest models, RF #1 (trained with $D_{forecast}$) and RF #2 (trained with $D_{tidy}$), tested on the winter seasons of 2018-2019 (top) and 2019-2020 (bottom) for predicting the danger levels forecasts. The blue shaded area represents the agreement rate and the points show the number of observers that provided an assessment. The dashed lines show the six days (labelled from 'a' to 'f') selected as exemplary cases. b) Box plots of the distribution of the accuracy of the models, grouped together by the agreement rate. Dots are the individual data points.

the agreement rate ranged between 27 % and 100 % (median 77 %, interquartile range 65 - 85 %), suggesting that the accuracy

of the forecast is lower than the overall model accuracy on about half of the days.

Fig. 7b summarizes the correlation between the daily prediction accuracy of the two RF models, evaluated against $D_{forecast}$, and the agreement rate between forecast and nowcast assessments. Again, we consider only days when at least 15 observers provided a nowcast assessment. Overall, the performance of both models decreased with decreasing agreement rate. When the





agreement was high ($P_\text{agree} > 0.9$ Fig. 7b), and hence the forecast in many places likely correct, the performance of RF #1 was
particularly good (median accuracy = 0.8), whereas the accuracy of RF #2 was slightly lower (median accuracy = 0.79). When
the agreement rate was low ($P_\text{agree} < 0.6$ Fig. 7b), and hence the forecast at least in some regions likely wrong, the predictive
performance of model RF #2, trained with the «tidy» danger level labels, is considerably lower, resulting in a median accuracy
of 0.66. In contrast, RF #1, which was trained with the over-forecast bias present in the $D_\text{forecast}$ data was less impacted (median
$\sim 0.7$).

**Exemplary case studies**

In order to highlight the variations in the daily accuracy, we selected six days that differed in terms of forecast accuracy
and model performance (Fig. 8). For simplicity, we only display the predictions of the model RF #1 (circles), for which we
additionally provide a video in the supplement. The maps of the predictions of the model RF #2 for these days are also available
in the supplement. Also shown in Fig. 8 are the local nowcast assessments for each of these six days (triangles).

On 7 February 2019 (Fig. 8a; denoted by 'a' in Fig. 7a), danger level 3-Considerable was forecast for most regions. For
this large area, the model predicted for the majority of the stations 2-Moderate, reaching a poor average daily accuracy of 0.3
(0.26 for RF #2). On this day, 27 observers provided a local assessment of the avalanche danger (Fig. 8a). Eight assessments
confirmed the forecast danger level 3-Considerable, 15 assessed the situation as 2-Moderate, suggesting that the forecast was
likely too high in many regions ($P_\text{agree} = 0.41$), and that the model actually performed well. In the remaining regions where the
forecast danger level was 2-Moderate, the observers mostly confirmed the forecast (one out of 4 reported 3-Considerable; Fig.
8a). The following day, the forecast danger level was lowered to 2-Moderate in almost all regions of the Swiss Alps.

On 15 March 2019 (Fig. 8b; denoted by 'b' in Fig. 7a), danger levels 3-Considerable and 4-High were mainly forecast. The
predictive accuracy on this day was 0.64 (0.62 for RF #2). Considering the local nowcast assessments showed that the danger
level forecast was perceived correct by 17 out of 23 observers ($P_\text{agree} = 0.74$). Five observers confirmed 4-High, five rated the
danger with 3-Considerable in the area where the forecast danger level was 4-High. In the regions with forecast danger level
3-Considerable, 12 observers confirmed the forecast danger level, one reported 2-Moderate (Fig. 8b).

On 19 March of 2019 (Fig. 8c; denoted by 'c' in Fig. 7a), danger levels 2-Moderate and 3-Considerable were forecast. For
rather large proportion of the stations in the area with 2-Moderate the model predicted one danger level higher, resulting in an
average model accuracy of 0.53. For RF #2, overall accuracy was considerable higher, namely 0.65. Fig. 8c shows that 79% of
the 24 local assessments on this day confirmed the forecast danger level; for 2-Moderate in 17 cases, for 3-Considerable for
2 out of 7 cases. This day seems to represent a typical example when RF #1, trained exclusively with forecast data, tended to
predict higher danger levels than RF #2.

On 20 January 2020 (Fig. 8d; denoted by 'd' in Fig. 7a), there were three areas with danger levels 1-Low, 2-Moderate and
3-Considerable, respectively. The average accuracy of the RF #1 model was 0.49, with many stations predicting a danger level
2-Moderate in the area where 1-Low was forecast. Two local assessments on this day confirmed 1-Low, eight 2-Moderate and
two 3-Considerable, while four observers in the area where 3-Considerable was forecast rated the danger as 2-Moderate, and
one as 1-Low in the area with 2-Moderate (Fig. 8d). In summary, this suggests that the forecast danger level was approximately





**Figure 8.** Maps of Switzerland showing the danger level of the public forecast for each region, the danger level predictions by RF#1 model at each IMIS station (coloured circles) and the local nowcast assessments (coloured triangles) reported by observers on six selected days: (a) 7 February 2019, (b) 15 March 2019, (c) 19 March 2019, (d) 20 January 2020, (e) 6 March 2020 and (f) 8 April 2020. The colours represent the danger levels.





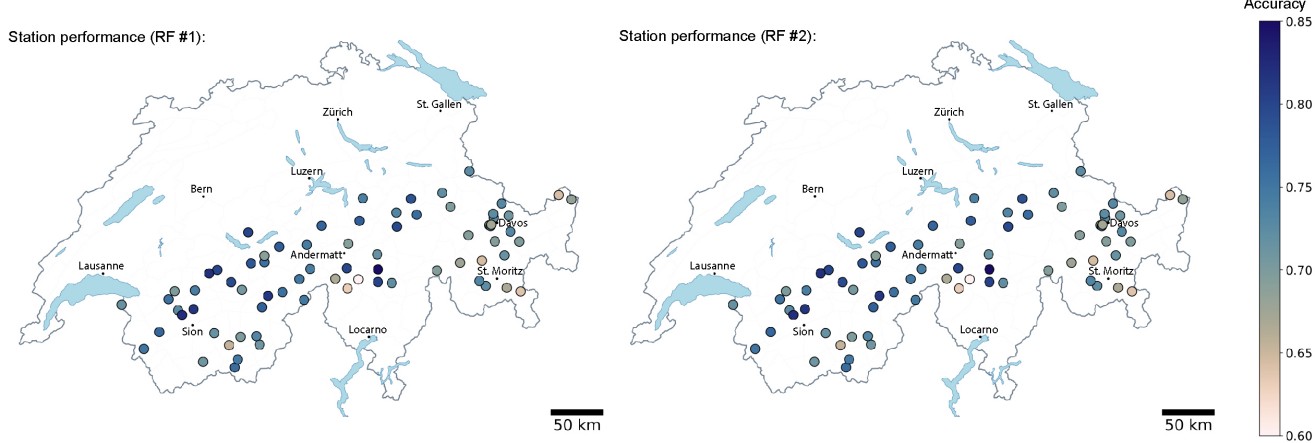

**Figure 9.** Maps showing the average accuracy of RF#1 and RF#2 model predictions for the 73 IMIS stations, for which predictions were available on at least 50% of the test set ($D_{\text{forecast}}$) days.

correct ($P_{\text{agree}} = 0.7$) but the model predictions tended to be too high, particularly in the area where 1-Low was forecast. The following day, the model predicted for most of the stations a decrease from 3-Considerable to 2-Moderate the following day, now again in accordance with the forecast. The performance of RF#2 was better (overall accuracy of 0.61) showing more accurate predictions in the large area where danger level 1-Low was forecast (see video in the supplementary material).

On 6 March 2020 (Fig. 8e; denoted by 'e' in Fig. 7a), when primarily danger level 3-Considerable was forecast, an accuracy of 0.81 was achieved by RF#1 (0.77 for RF#2). However, the feedback from the observers (Fig. 8e), with 15 out of the 27 local assessments being lower than the forecast danger level, suggests that the forecast danger level was at least in some regions too high ($P_{\text{agree}} = 0.46$). Similarly, the avalanche observations indicated only for one warning region that level 4-High was appropriate.

Finally, on 8 April 2020 (Fig. 8f; denoted by 'f' in Fig. 7a), the lowest danger level 1-Low was forecast for the entire area of the Swiss Alps. Both models also predicted 1-Low for all stations, an accuracy of 1. On this day, only four observers provided a local nowcast estimate, all of which were in accordance with the forecast danger level (Fig. 8f).

## 5.3 Station-specific model performance

Our objective was to develop a generally applicable classifier for predicting the danger level at all IMIS stations in the Swiss Alps. In other words, the classifier should show a similar performance independent of the location of the station. To explore this, we analyzed the station-specific averaged accuracy for the entire test set ($D_{\text{forecast}}$) of both models for the 73 stations, for which predictions were available in at least 50% of the days.

The maps displayed in Fig. 9 show that the station-specific accuracies of RF#1 ranged between 0.6 and 0.85 (mean accuracy = 0.73) and between 0.5 and 0.87 (mean accuracy = 0.72) for RF#2. Some spatial patterns in the performance of both models are visible (Fig. 9), indicating that differences between stations are not random: both models performed consistently well in





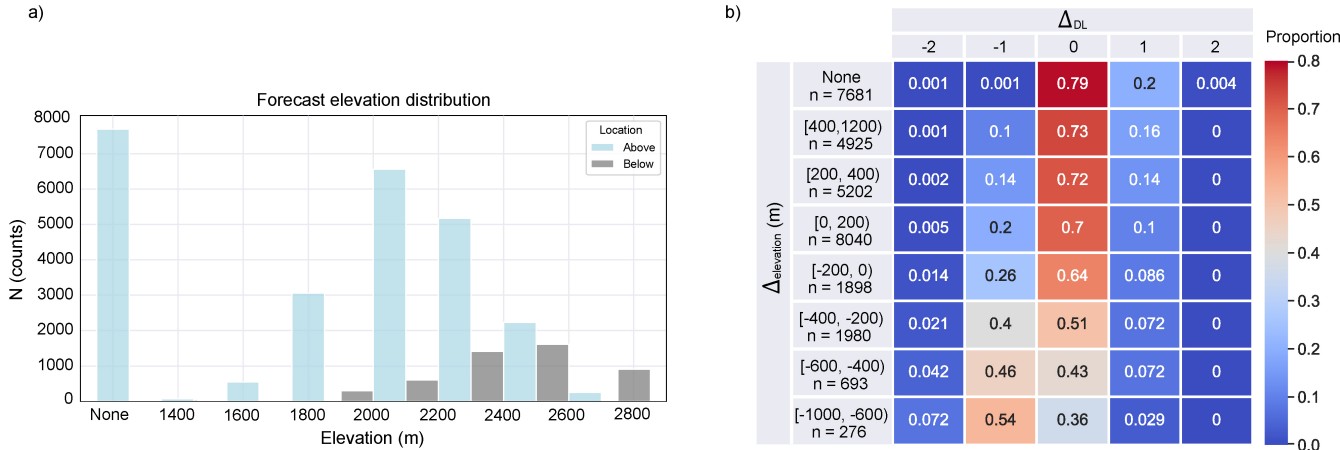

**Figure 10.** a) Frequency of the elevation indicated in the public forecasts with the number of stations that are located above and below this elevation. The class 'none' contains the samples for the days when no information was indicated in the bulletin. b) Heat map of the proportions of samples (row-wise normalized) for each of the eight elevation classes ($\Delta_{elevation}$) versus the range of prediction bias ($\Delta_{DL}$) of the model RF #1. The total number of samples in each elevation class is denoted with 'n'.

the northern and western parts of the Swiss Alps with the accuracy being above the mean for many stations, compared to much lower accuracy in the eastern part of the Alps (accuracy $< 0.7$). RF #1 performed somewhat better in the southern and central parts of Switzerland and RF #2 in the northern parts. At stations with a lower performance (accuracy $< 0.7$), the mean proportion of days forecast when danger levels 1-Low or 3-Considerable were forecast was slightly lower ($\sim 3\%$). As the prediction performance was higher at these danger levels (Table 1a and b), this may partly explain the geographical differences in performance.

**5.4 Model performance with elevation**

Here we address the impact of filtering for elevation, which we applied for data preparation when defining the training and test data. We trained the classifiers exclusively with data from stations which were above the elevation indicated in the bulletin (Sect. 3.3, see also Fig. 2). To explore whether this decision was appropriate, we now compare the prediction accuracy of RF #1 as a function of the difference in elevation between the stations and the elevation indicated in the bulletin: $\Delta_{elevation} =$ elevation(station)-elevation(forecast).

In the public bulletin, the elevation information is given in increment intervals of 200 m in the range between 1400 and 2800 m a.s.l. for dry-snow conditions. To get more insight into the performance of the model in relation to the elevation, we separated the predictions into those for stations located above ($N = 25.848$) and those below ($N = 4847$) the elevation indicated in the bulletin (Fig. 10a). Generally, on any given day, the elevation indicated in the forecast is lower than the elevation of most stations.




**Table 3.** Accuracy of RF classifier tested against $D_{\text{forecast}}$ as a function of changes in the forecast danger level compared to the day before, for cases when the danger level increased ($\nearrow$), stayed the same ($\rightarrow$), or decreased ($\searrow$) for (a) RF #1, and (b) for RF #1*, a model which additionally considers the forecast danger level of the previous day as an input feature.

| Danger level | a) RF #1 | | | | b) RF #1* | | | |
|---|---|---|---|---|---|---|---|---|
| | $\nearrow$ | $\rightarrow$ | $\searrow$ | All | $\nearrow$ | $\rightarrow$ | $\searrow$ | All |
| 1-Low | - | 0.83 | 0.54 | 0.79 | - | 1 | 0.17 | 0.88 |
| 2-Moderate | 0.57 | 0.70 | 0.53 | 0.66 | 0.20 | 0.96 | 0.21 | 0.76 |
| 3-Considerable | 0.74 | 0.80 | 0.93 | 0.79 | 0.50 | 0.95 | 0.86 | 0.85 |
| 4-High | 0.48 | 0.55 | - | 0.51 | 0.42 | 0.78 | - | 0.55 |
| Overall | 0.67 | 0.77 | 0.59 | 0.74 | 0.43 | 0.97 | 0.29 | 0.82 |

To analyze the model performance in more detail, we defined eight classes of $\Delta_{elevation}$. Fig. 10b shows the eight classes and their definitions, each containing the proportion of samples (row-wise sum) as a function of the model bias difference defined in Equ. 1. The class 'none' contains the samples for the days, when no elevation information was provided in the bulletin. This class essentially corresponds to forecasts with danger level 1-Low (99 %). This is the most accurate class, reaching an accuracy at $\Delta_{DL} = 0$ of 79 %, which is the same as the recall for 1-Low shown in Table 1a. Overall, the prediction accuracy was highest for stations with an elevation far above the elevation indicated in the forecast (accuracy 0.73 for $\Delta_{\text{elevation}} \geq 400$), and lowest for stations located far below this elevation (accuracy 0.36 for $\Delta_{\text{elevation}} \leq -600$, Fig. 10b). At the same time, the bias in the predictions, compared to $D_{\text{forecast}}$, changed from being slightly positive (ratio of the proportion of predictions higher vs. lower than forecast: 1.6 for $\Delta_{\text{elevation}} \geq 400$) to negative ($\Delta_{\text{elevation}} \leq 200$), and to primarily being negative for stations far below this elevation (ratio predictions lower vs. higher than forecast: 18 for $\Delta_{\text{elevation}} \leq -600$, Fig. 10b).

## 5.5 Model performance with respect to increasing or decreasing hazard

When evaluating the accuracy of the avalanche forecast, Techel and Schweizer (2017) distinguished between days when the avalanche danger increased and days when it decreased. When the danger increases, primarily changing weather drives the decrease in snow stability. In contrast, decreasing avalanche danger is often linked to comparably minor and/or slow changes in snowpack stability (e.g. Techel et al., 2020b). While these changes are gradual in nature, these can only be expressed in a step-like fashion using the five-level danger scale. For the purpose of this analysis, we followed the approach by Techel and Schweizer (2017), and split the data set into days when the danger level increased, stayed the same, or decreased, in relation to the previous day.

As shown in Table 3a, the accuracy was highest on days when the forecast danger level stayed the same (0.77), compared to days when the forecast danger increased (accuracy = 0.67; support = 10 %) or decreased (accuracy = 0.59; support = 12 %). Considering that Techel and Schweizer (2017) reported the lowest agreement between forecast and nowcast for days when the forecast increased, suggests that we evaluate these cases with danger level labels which were proportionally more often wrong.

 

## 5.6 Model performance considering the forecast danger level from previous day

The avalanche warning service daily reviews the past forecast in the process of preparing the future forecast (Techel and Schweizer, 2017). Hence, the past forecast can be seen as the starting point for the future forecast. Therefore, we also tested whether the prediction performance changed when including the forecast danger level from the previous day's forecast as an additional feature in the random forest model (RF#1*). As shown in Table 3b, the overall accuracy increased notably from 0.74 (RF#1) to 0.82 (RF#1*), but also for all the danger levels individually. However, when additionally considering the change to the previous day's forecast, this comes at the cost of a large decrease in the performance in situations when the danger level changed ($DL$ increased on 10 % and decreased on 12 % of the total samples). For these situations, there is a drop in accuracy, overall from 0.67 (RF#1) to 0.43 (RF#1*) when the danger level increased, and from 0.59 (RF#1) to 0.29 (RF#1*) when the danger level decreased.

## 6 Discussion

We trained two random forest classifiers, RF#1 and RF#2, building on two different ground truth data sets. The first data set, $D_{\text{forecast}}$, relied purely on the information provided in the public bulletin and is expected - in terms of the quality of the labels - to be intrinsically noisier (Sect. 3.1.1). Given the uncertainty related to the labels of $D_{\text{forecast}}$, we compiled a second data set, $D_{\text{tidy}}$, by combining the information provided in the forecast with additional pieces of evidence to obtain a target variable with, presumably, more reliable labels (Sect. 3.1.2 and Appendix A).

We first discuss key characteristics of the training data (Sect. 6.1), which may impact both the construction of the RF classifiers and their performance evaluation:

– the size of the data set in relation to the complexity of the addressed classification problem,

– the class distribution, with particular attention to minority classes, and

– the quality of the labels.

Furthermore, of particular relevance are the discrete nature of the target variable, a strong over-simplification of the continuous phenomenon of avalanche danger (Sect. 6.2) and scale issues - a danger level describing regional avalanche conditions for a whole day compared to measurements and SNOWPACK simulation output describing a specific point in time and space (Sect. 6.3). Finally, we discuss the performance of the RF classifiers considering one of our key objectives, namely to develop a model, applicable to the entire forecast domain of the Swiss Alps (Sect. 6.4), before we compare the developed RF classifiers with previously developed models predicting a regional avalanche danger level (Sect. 6.5).





## 6.1 Impact of training data and labelling in model performance

**Training data size and class distribution**

In general, a large training data set increases the performance of a machine learning model as it provides more coverage of the data domain. However, Rodriguez-Galiano et al. (2012) showed that random forest classifiers have relatively low sensitivity to the reduction of the size of the training data set. In fact, the large reduction in the amount of training data of RF #2, containing only the 10 % of data of RF #1, did not have a substantial impact on model performance. RF #2 had similar overall scores when evaluated on the $D_{\text{forecast}}$ test set (Table 1a and b) and even slightly higher scores on the $D_{\text{tidy}}$ test set (Table 1c and d). The dominant classes of danger levels, 2-Moderate and 3-Considerable, were the most affected ones, showing a decrease in accuracy between 5 % and 8 % (Fig. 6a and b).

In addition, imbalanced classification poses a challenge for predictive modelling as most existing classification algorithms such as random forests were designed assuming an equal class distribution of the training set, giving rise to lower accuracy for minority classes (Chen et al., 2004). Since danger level 5-Very High is very rarely forecast (< 0.1 %), we merged it with 4-High. This step reduced the multi-class classification problem to four classes. We also explored diverse data sampling techniques (results not shown), down-sampling the majority classes or over-sampling the minority classes, to balance the training data when fitting random forest. However, since none of these methods showed an improvement in the performance and given the imbalanced nature of the avalanche forecasting domain, we discarded these strategies. Hence, we opted for learning from our extremely imbalanced data set applying cost-sensitive learning that assigned larger weights to minority classes (Sect. 4).

Furthermore, RF #2 is trained using a better-balanced training data set (Fig. 3c). The confusion matrices exhibit an improvement of the per class accuracy (Fig. 6), i.e. the recall percentages of the diagonal matrix, of the minority classes of danger 1-Low and 4-High when using RF #2, reflecting the positive impact of balancing the training ratio for this danger level.

**Quality of the labels**

Incorrect labels in the $D_{\text{forecast}}$ data set are inherent in our problem domain as avalanche forecasts are sometimes wrong due to inaccurate weather forecasts, variations in local weather and snowpack conditions and human biases and noise (McClung and Schaerer, 2006). In terms of the target variable, these errors may manifest themselves in errors in the danger level, the elevation information indicated in the forecast, or the spatial extension of regions with a specific danger level. Furthermore, all of these elements are gradual in nature, and not step-like as the danger level, the elevation band and the delineation of the warning regions would suggest. Even though previous applications of random forests have demonstrated that it is one of the most robust classification methods tolerating some degree of label noise (e.g. Pelletier et al., 2017; Frénay and Verleysen, 2013), its performance decreases with a large amount of label errors (Maas et al., 2016). Labelling errors, however, may influence the model building, which can be particularly relevant for minority classes as danger level 4-High. Furthermore, such errors in the ground-truth may also lead to seemingly lower prediction performance (e.g. Bowler, 2006; Techel, 2020). Aiming to reduce the impact of wrong class labels, we compiled the best possible, and presumably more accurate, ground truth data set, ($D_{\text{tidy}}$), which was used to train RF #2.



To assess the accuracy of the forecast, and thus potential errors in the forecast danger levels ($D_{\text{forecast}}$), we relied on nowcast assessments ($DL_N$) by well-trained observers. Although the local nowcasts are also subjective assessments, they are considered the most reliable data source of danger levels (Schweizer et al., 2021; Techel and Schweizer, 2017). Previous studies estimated the accuracy of the Swiss avalanche forecasts in the range between 75 % and 81 % (this study, see Sect. 5.2; Techel and Schweizer, 2017; Techel et al., 2020b). Our classifiers reached these values: the overall prediction accuracies of RF#1 and RF#2 were 74 % and 72 % (compared to $D_{\text{forecast}}$) and, 76 % and 78 % (compared to $D_{\text{tidy}}$), respectively (Table 1). Particularly, the accuracy of the minority class 4-High improved for RF#2 (Fig. 6), emphasizing the importance of training and testing against the best possible data set $D_{\text{tidy}}$. To compile this data set, quality checking was particularly important for danger level 4-High (Sect. 3.1.2 and Appendix Sect. A), since the forecast is known to be comparably often erroneous when this danger level is forecast (e.g. Techel and Schweizer, 2017; Techel, 2020). In the future, a new compilation of $D_{\text{tidy}}$ resulting in a larger data size may improve the predictive performance.

Considering the predictions on particular days (Fig. 8), some stations predicted the danger level, which was forecast in the adjacent warning region. This suggests that occasionally the boundary between areas of different forecast danger level could be questionable. Such errors in the spatial delineation of the extent of regions with the same danger level have also been noted by Techel and Schweizer (2017). They showed that the agreement rate between the local nowcast assessments and the regional forecast danger level was comparably low in warning regions which were neighbours to warning regions with a different forecast danger level. Hence, incorrect boundaries may have further contributed to label noise.

Similarly, errors in the elevation indicated in the bulletin may have an impact, as we used this forecast elevation to filter data (Sect. 3.3). The effect of the forecast elevation on the classifier performance was clearly visible with the accuracy decreasing for stations below the elevation indicated in the bulletin, often showing a bias of -1 danger level (Fig. 10). This result agrees with the assumption that the danger is lower below the elevation indicated, typically by one danger level (Winkler et al., 2021). However, the proportion of correct predictions at stations close, but below the elevation indicated was fairly high (0.64), which may reflect a more gradual decrease in the danger level with elevation (Schweizer et al., 2003). This finding suggests that the model is able to capture elevational gradients in avalanche danger.

## 6.2 Avalanche danger levels - a discrete description of a continuous phenomenon

The avalanche danger levels are a strong simplification of avalanche danger, which is a continuous phenomenon. However, the random forest classifiers not only predict the most likely danger level, which we exclusively explored in this study, but also the class probabilities for each of the danger levels. Even though an in-depth analysis of these probabilities is beyond the scope of this study, we noted that for most of the misclassifications between two consecutive danger levels (Fig. 8), the model predictions usually were uncertain, predicting high probabilities of belonging to both danger levels. In the future, using these probability values may be beneficial for refining the avalanche forecasts (Techel et al., 2020b).




### 6.3 Spatio-temporal scale issues

The temporal and spatial scale of the avalanche forecast and data used to train the model should be considered when verifying
a forecasting model (McClung, 2000). To match the temporal scale, we extracted the meteorological and snowpack features
for the time window closest to the avalanche forecast. Nevertheless, for avalanche forecasting, 'forecast' data from weather
predictions strongly drives the decision-making process. The RF models, however, were trained using 'nowcast' data (recorded
measurements and simulated data based on these measurements). This may introduce an additional bias between the danger
level predictions of the model and the public forecast. The use of the morning forecast, whenever it was available as ground
truth, reduced this bias. Nevertheless, a model trained with 'forecast' input data may improve the performance.

A scale mismatch exists between our target variable and the model predictions. Whereas the same danger level is usually
issued for a cluster of warning regions, characterized by a mean size of 7000 km² (Techel and Schweizer, 2017), the predictions
of the model reflect the local conditions measured and modelled at an individual IMIS station. Hence, the spatial scale differ-
ence can be more than two orders of magnitude. Stations located in the same or nearby warning regions forecast with the same
danger level sometimes predict different danger levels (Fig. 8) as avalanche conditions may vary even at the scale of a warning
region (Schweizer et al., 2003). These local variations are inherent to the characteristics of the station such as elevation, wind
exposure and more. To overcome the spatial scale issue, predictions could be clustered through ensemble forecasting methods.
However, in some areas, the number of stations is insufficient for such an approach (Fig. 1).

### 6.4 Spatio-temporal variations of the model performance

Snow stability, and hence avalanche danger, evolves in time - driven primarily by changing weather conditions, and varies in
space - depending on the terrain and how meteorological conditions affect the snowpack at specific locations.

Overall, the two models captured this evolution with an overall accuracy of more than 72 % (Tab. 1), or 67 % (RF#1) when
considering only times when the avalanche hazard increased (Tab. 3a). However, the accuracy of the models varied during the
winter season (Fig. 7a), with about 10 - 15 % of the days exhibiting an accuracy $< 0.5$ (Sect. 5.1). Here, we distinguished two
cases (Sect. 5.2): firstly, some days with such seemingly poor performance could be linked to the forecast danger level, the
target variable used for validation, likely being wrong in many areas. These cases were characterized by a low agreement rate,
$P_{\mathrm{agree}}$, between forecast and nowcast assessments, as for instance on 7 February 2019 (Fig. 8a). However, not all the days with a
poor model performance correlated with low values of $P_{\mathrm{agree}}$ (Fig. 7b). This suggests that variations in model performance may
also be due to different avalanche situations and, hence, the ability of the classifiers to accurately predict them. Even though
we have only qualitatively explored this, we observed that the predictive performance of both models sometimes decreased on
days when the avalanche problem 'persistent weak layers' (EAWS, 2021) was the primary problem.

The performance of the models was lower at stations located in the eastern part of the Swiss Alps, as for instance, in the
regions surrounding Davos or St. Moritz (these are marked in Fig. 9). Since model accuracy varied in situations when the
danger changed (Table 3), we verified whether the proportion of cases with a change in the danger level differed in these
regions compared to other areas. However, changing danger levels were about as often forecast in these regions as in the rest




of Switzerland, with, for instance, an increase in avalanche danger being forecast on 9 to 10 % of the days in Davos and St. Moritz, compared to an overall mean of 10 % for the remainder of the Swiss Alps (decreasing danger level: 11 % to 12 % in St. Moritz and Davos, respectively, overall mean 13 %). The model performance was highest when danger level 1-Low was forecast (Tab. 1), which was somewhat less frequently the case in St. Moritz (24 %) and Davos (26 %) compared to the entire

Swiss Alps (29 % top of Fig. 3d). Furthermore, we also explored if the agreement rate between forecast and local assessments, an indicator for the quality of the danger level labels, was lower there. While $P_{\mathrm{agree}}$ was about 71 % for Davos, which was lower than the overall mean of 75 %, the agreement rate was 82 % for St. Moritz. Consequently, none of these effects may conclusively explain the variations observed. However, again a possible explanation may be related to the snowpack structure in this part of the Swiss Alps, which is often dominated by the presence of persistent weak layers (e.g. Techel et al., 2015).

However, this aspect of model performance must be analyzed in more detail and goes beyond the scope of this work.

### 6.5 Comparison of data-driven approaches for danger level predictions

Some of the first attempts to automatically predict danger levels were reported by Schweizer et al. (1994), who designed a hybrid expert system based on a training set of about 700 cases using a verified danger level, correctly classifying 73 % of the cases. Schweizer and Föhn (1996) also predicted the avalanche danger level for the region of Davos trained with the same data

that included two variables related to snowpack stability. The cross-validated accuracy was 63 %, showing an improvement to 73 % when adding further snowpack stability data and knowledge in the form of expert rules to the system.

Schirmer et al. (2009) compared several classical machine learning methods such as classification trees, Hidden Markov models, recurrent artificial neural networks, and nearest-neighbor methods to predict the avalanche danger. They used as input measured meteorological and SNOWPACK variables from the AWS at Weissfluhjoch (WFJ2) station located above Davos.

They reported an accuracy typically around 55 % to 60 %, which improved to 73 % when the avalanche danger level of the previous day was an additional input. Although the test set used in this study is not directly comparable with the previous ones, the overall accuraccies obtained with our classifiers are higher (Table 1). Still, the mean accuracy of the predictions at the the stations located in the region of Davos was lower (Fig. 9), showing values of 72 % (RF#1 model) and 69 % (RF#2 model) for the station WFJ2. We also observed an important improvement of the overall performance of the model when adding the danger

level of the previous day (Table 3b). However, the predictions were mainly driven by the danger level feature and RF#1* failed to predict the situations of increasing or decreasing avalanche hazard. This model would have limited usefulness in operational avalanche forecasting, since it too strongly favors persistency in avalanche danger.

### 7 Conclusions

We developed two random forests classifiers to predict the avalanche danger level based on data provided by a network of

automated weather stations in the Swiss Alps (Fig. 1). The classifiers were trained using measured meteorological data and the output of snow cover simulations driven with these input weather data, and danger ratings from public forecasts as ground truth. The first classifier RF#1 relied on the actual danger levels as forecast in the public bulletin, $D_{\mathrm{forecast}}$, which is intrinsically


noisy, while the second classifier RF #2 was labelled with a subset of quality-controlled danger levels, $D_{\text{tidy}}$. Whereas, for the classifier RF #1 the maximum average accuracy ranged between 74 % (evaluating on $D_{\text{forecast}}$ test set) and 76 % ($D_{\text{tidy}}$ test set);

RF #2 showed an accuracy between 72 % ($D_{\text{forecast}}$ test set) and 78 % ($D_{\text{tidy}}$ test set). These accuracies were higher than those obtained in earlier attempts of predicting the danger level. Also, our classifiers had similar accuracy as the Swiss avalanche forecasts, which were estimated by Techel and Schweizer (2017) in the range of 70-85 % with an average value of 76 %. Hence, we developed a fully data-driven approach to automatically predict avalanche danger with an accuracy comparable to the experience-based avalanche forecasts in Switzerland. Overall, the performance of the RF models decreased with increasing

uncertainty related to these forecasts, i.e. decreasing agreement rate ($P_{\text{agree}}$). In addition, the predictions at stations located at elevations higher than the elevation indicated in the bulletin were more accurate than the predictions at lower stations, suggesting, as expected, lower danger at elevations below the critical elevations. Finally, a single model was applicable to the different snow climate regions that characterize the Swiss Alps. Nevertheless, the predictive performance of the models spatially varied and in some eastern parts of the Swiss Alps where the avalanche situation is often characterized by the presence of persistent

weak layers, the overall accuracy was lower ($\sim 70\%$). Therefore, future models should better address this particular avalanche problem by incorporating improved snow instability information.

Both models have the potential to be used as a decision support tool for avalanche forecasters when assessing avalanche hazard. Operational pre-testing during the winter season 2020-2021 already showed promising results for the real application in operational forecasting. Future work will focus on exploiting the output probabilities of the random forest classifiers, predicting

the danger levels for the different slope aspects in addition to using output of numerical weather prediction models as input data. These future developments would bring the models even closer to the procedures of operational avalanche forecasting.

*Author contributions.* CP: concept and design of study, data collection and curation, model development, computational framework, analysis, writing, FT: concept and design of study, data collection and curation, analysis, writing, reviewing, MH: concept and design of study, data collection and curation, model development, analysis, reviewing, MV: data curation, model development, computational framework, analysis,

writing, reviewing, AH: concept and design of study, model development, reviewing, TO: computational framework, reviewing, GO: model development, reviewing, FP: model development, reviewing, JS: concept and design of study, model development, writing, reviewing.

*Competing interests.* No competing interests are present.

*Video supplement.* For illustration, the evolution of the RF danger level predictions (circles), the local nowcast assessments (circles) and the forecast danger level ($D_{\text{forecast}}$, Fig. 3d) is shown for the two test winters in two supplementary videos. Each video shows animations of the

daily maps. Only the predictions for stations above the elevation indicated in the bulletin are displayed. The warning regions are colored with the forecast danger level. The color of the stations shows the danger level predictions of each random forest classifier. The number of stations varies with time because predictions on some stations are lacking due to i) the station was located on a given day below the elevation



indicated in the bulletin, ii) a missing value for one of the input features or iii) the snow height was less than the minimum threshold of 30 cm. The danger level of some warning regions can also be missing for some days because only a forecast for wet-snow avalanche conditions was issued in this area.

*Acknowledgements.* Marc Ruesch and Mathias Bavay provided access to avalanche forecast and weather station data, and the SNOWPACK simulations. This study was funded by the collaborative data science project scheme of the Swiss Data Science Center.





**Appendix A:  Compilation of subset of «tidy» danger levels ($D_{\text{tidy}}$)**

In the following, the data and process to obtain the subset of «tidy» danger levels, introduced in Sect. 3.1.2, are described.
Several data sources were used:

1. the forecast danger level ($D_{\text{forecast}}$) relating to dry-snow conditions, as described in Sect. 3.1.1;

2. nowcast estimates of the danger level ($D_{\text{nowcast}}$) relating to dry-snow conditions, and reported by experienced observers after a day in the field (refer to Techel and Schweizer (2017) for details regarding nowcast assessments of avalanche danger in Switzerland);

3. avalanche occurrence data, consisting of recordings of individual avalanches and avalanche summaries, reported by the observer network in Switzerland for the purpose of avalanche forecasting;

4. «verified» danger levels, as shown in studies exploring snowpack stability in the region of Davos (eastern Swiss Alps, see also Fig. 2,  Schweizer et al., 2003; Schweizer, 2007) or documenting avalanche activity following two major storm in 2018 and 2019 using satellite-detected avalanches (Bühler et al., 2019; Bründl et al., 2019; Zweifel et al., 2019).

We proceeded in two steps to derive $D_{\text{tidy}}$.

(1) We combined information provided in the forecast ($D_{\text{forecast}}$) with assessments of avalanche danger by observers ($D_{\text{nowcast}}$). By combining several pieces of information indicating the same $D$, we expect that it is more likely that $D$ represents the avalanche conditions well. This resulted primarily in a subset of danger levels 1-Low, 2-Moderate and 3-Considerable. We included the following cases in the tidy subset:

– For cases, when a single nowcast estimate was available, and when $D_{\text{forecast}} = D_{\text{nowcast}} \rightarrow D_{\text{tidy}} = D_{\text{forecast}}$;

– For cases, when several nowcast estimates were available, and when these indicated the same $D_{\text{nowcast}}$, regardless of $D_{\text{forecast}} \rightarrow D_{\text{tidy}} = D_{\text{nowcast}}$.

Furthermore, we included cases, when a «verified» danger level was available (Schweizer et al., 2003; Schweizer, 2007). When neither a «verified» danger level nor a nowcast estimate was available, but when $D_{\text{forecast}}$ was 1-Low on the day of interest, but
also on the day before and after, we included these cases as sufficiently reliable to represent 1-Low. However, to reduce auto-correlation in this subset of days with 1-Low, only every fifth day was selected. Furthermore, as our focus was on dry-snow conditions, we removed all cases of 1-Low in April, when often a decrease in snow stability during the day due to melting leads to a wet-snow avalanche problem.

Beside compiling $D_{\text{tidy}}$, we also derived a corresponding critical elevation and aspects, for which $D_{\text{tidy}}$ was valid.

We defined a «tidy» critical elevation as the mean of the indicated elevations in the forecast or nowcast estimates. As generally no elevation is provided for 1-Low in the forecast nor in nowcast assessments, we used a fixed elevation of 1500 m for the months December to February, and 2000 m in March. The latter adjustment was made to ascertain that the danger referred to





dry-snow avalanche conditions rather than wet-snow or gliding avalanche conditions.

(2) We relied on avalanche occurrence data to obtain a subset of cases, which reflect the two higher danger levels 4-High and 5-Very High.

To find days with avalanche activity typical for danger level 4-High, an avalanche activity index (AAI) was calculated for each day and warning region by summing up the number of reported avalanches weighted according to their size (Schweizer et al., 1998). The respective weights for avalanche size classes 1 to 4 were: 0.01, 0.1, 1, 10. Because a mix of individual avalanche recordings and avalanche summary information was used, the following filters and weights were applied to calculate the AAI:

- Individual avalanche recordings: only dry-snow natural avalanches were considered (weight = 1).

- Avalanche summaries: only avalanches classified as either dry (weight = 1) or a mix of dry and wet (weight = 0.5), which had released either naturally (weight = 1) or were reported as a mix of natural and other release types (weight = 0.5) were used.

A day and warning region was considered as 4-High, when the following three criteria were fulfilled:

1. At least one avalanche was of size 3, or larger.

2. AAI $\geq$ 5. This threshold corresponds to, for example, five natural avalanches of size 3, or forty size 2 avalanches and one size 3 avalanche.

3. At least five avalanches of size 2, or larger, were reported.

Cases, which fulfilled these criteria, were included and $D_{\text{tidy}}$ was set to 4-High if $D_{\text{forecast}}$ was $\geq$ 3-Considerable. Cases, for which the avalanche activity criteria were fulfilled but which had a comparably low danger level forecast ($D_{\text{forecast}}$ = 1-Low or 2-Moderate), were removed from the subset.

5-Very High: Two situations were verified as 5-Very High for parts of the Swiss Alps - 22 Jan 2018 (Bründl et al., 2019) and 14 Jan 2019 (Zweifel et al., 2019). These cases were included in the data set. If one of the previous criteria already applied, $D_{\text{tidy}}$ was changed to 5-Very High.

For cases with $D \geq$ 4-High, which did not contain information on elevation, we used a rounded mean based on the cases where this information was available. This resulted in a critical elevation of 1900 m.

**Appendix B: Metrics and definition of features used for random forest**

In the following, the performance metrics used in this study are defined (e.g. Sokolova and Lapalme, 2009). The accuracy is the fraction of predictions by the model that are correct:

$$Accuracy = \frac{Correct\ predictions}{Total\ predictions} \tag{B1}$$





Precision (or positive predictive value) describes the fraction of positive results that are true positives:

$$Precision = \frac{TruePositive}{TruePositive + FalsePositive} \tag{B2}$$

Recall describes the true positive rate (or sensitivity), i.e. the percentage of actual positives which are correctly identified:

$$Recall = \frac{TruePositive}{TruePositive + FalseNegative} \tag{B3}$$

The F1 score is the harmonic mean of precision and recall:

$$F1 = 2 * \frac{precision * recall}{precision + recall} \tag{B4}$$

Macro-F1 is the unweighted mean of F1 scores calculated for each class.





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





**Table B1.** Meteorological variables used for training the random forest algorithm. The three types of features are: measured meteorological variable, modelled meteorological variable by SNOWPACK or extracted variable. Features can be discarded by Recursive Feature Elimination (RFE), manually or because they are highly correlated with another one.

| Feature description | Feature name | Type | Selected/discarded |
|---|---|---|---|
| Mean sensible heat [W/m$^2$] | Qs | Modelled | Selected |
| Mean latent heat [W/m$^2$] | Ql | Modelled | Discarded: RFE |
| Mean ground temperature [°] | TSG | Measured | Discarded: RFE |
| Mean ground heat at soil interface [W/m$^2$] | Qg0 | Modelled | Selected |
| Mean rain energy [W/m$^2$] | Qr_mean | Modelled | Discarded: correlation |
| Mean outgoing long wave radiation [W/m$^2$] | OLWR | Modelled | Discarded: correlation |
| Mean incoming long wave radiation [W/m$^2$] | ILWR | Modelled | Selected |
| Mean net long wave radiation [W/m$^2$] | LWR_net | Modelled | Selected |
| Mean reflected short wave radiation [W/m$^2$] | OSWR | Measured | Discarded: correlation |
| Mean incoming short wave radiation [W/m$^2$] | ISWR | Modelled | Selected |
| Mean net short wave radiation [W/m$^2$] | Qw | Modelled | Selected |
| Mean parametrized albedo [−] | pAlbedo | Modelled | Selected |
| Mean incoming short wave on horizontal [W/m$^2$] | ISWR_h | Modelled | Discarded: correlation |
| Mean direct incoming short wave [W/m$^2$] | ISWR_dir | Modelled | Discarded: correlation |
| Mean diffuse incoming short wave [W/m$^2$] | ISWR_diff | Modelled | Selected |
| Mean air temperature [°] | TA_mean | Measured | Selected |
| Mean surface temperature [°] | TSS_mod | Modelled | Selected |
| Mean surface temperature [°] | TSS_meas | Measured | Discarded: correlation |
| Mean bottom temperature [°] | T_bottom_meas | Modelled | Discarded: correlation |
| Mean relative humidity [−] | RH | Measured | Selected |
| Mean wind velocity [m/s] | VW | Measured | Selected |
| Mean wind velocity drift [m/s] | VW_drift | Measured | Selected |
| Mean wind direction [°] | DW | Measured | Discarded: RFE |
| Mean solid precipitation rate [kg/s$^2$/h] | MS_Snow | Modelled | Selected |
| Mean snow height [cm] | HS_mod | Modelled | Selected |
| Mean snow height [cm] | HS_meas | Measured | Discarded: correlation |
| Mean hoar size [cm] | hoar_size_meas | Modelled | Discarded: RFE |
| Mean 24h wind drift [cm] | wind_trans24 | Modelled | Selected |
| Mean 3d wind drift [cm] | wind_trans24_3d | Extracted | Selected |
| Mean 7d wind drift [cm] | wind_trans24_7d | Extracted | Selected |
| Mean 24h height of new snow [cm] | HN24 | Modelled | Selected |
| Mean 3d sum of daily height of new snow [cm] | HN72_24 | Modelled | Selected |
| Mean 7d sum of daily height of new snow [cm] | HN24_7d | Extracted | Selected |
| Mean snow water equivalent [kg/m$^2$] | SWE | Modelled | Discarded: correlation |
| Mean total amount of water [kg/m$^2$] | MS_Water | Modelled | Discarded: RFE |
| Mean erosion mass loss [kg/m$^2$] | MS_Wind | Modelled | Discarded: RFE |
| Mean rain rate [kg/s$^2$/h] | MS_Rain | Modelled | Discarded: correlation |
| Mean virtual lysimeter [kg/s$^2$/h] | MS_SN_Runoff | Modelled | Discarded: RFE |
| Mean sublimation mass [kg/m$^2$] | MS_Sublimation | Modelled | Discarded: correlation |
| Mean evaporated mass [kg/m$^2$] | MS_Evap | Modelled | Discarded: RFE |
| Mean snow temperature at 0.25 m [°] | TS0 | Modelled | Discarded: manually |
| Mean snow temperature at 0.5 m [°] | TS1 | Modelled | Discarded: manually |
| Mean snow temperature at 1 m [°] | TS2 | Modelled | Discarded: manually |
| Mean stability class [-] | Sclass2 | Modelled | Discarded: RFE |
| Mean deformation rate stability index [−] | Sd | Modelled | Discarded: RFE |
| Mean depth of deformation rate stability index [cm] | zSd | Modelled | Discarded: correlation |
| Mean natural stability index [−] | Sn | Modelled | Selected |
| Mean depth of natural stability index [cm] | zSn | Modelled | Selected |
| Mean Sk38 skier stability index [−] | Ss | Modelled | Selected |
| Mean depth of Sk38 skier stability index [cm] | zSs | Modelled | Selected |
| Mean structural stability index [−] | S4 | Modelled | Selected |
| Mean depth of structural stability index [cm] | zS4 | Modelled | Discarded: correlation |
| Mean stability index 5 [−] | S5 | Modelled | Discarded: RFE |
| Mean depth of stability index 5 [cm] | zS5 | Modelled | Discarded: RFE |




**Table B2.** Variables extracted from the simulated profiles used for training the random forest algorithm. Features can be discarded by Recursive Feature Elimination (RFE) or because they are highly correlated with another one.

| Feature description | Feature name | Type | Selected/discarded |
|---|---|---|---|
| Persistent weak layer(s) in the 100 cm from the surface [−] | pwl_100 | Profile | Discarded: correlation |
| Persistent weak layer(s) at depths between 15 cm and 100 cm [−] | pwl_100_15 | Profile | Discarded: correlation |
| Persistent weak layer at bottom [−] | base_pwl | Profile | Discarded: RFE |
| Structural stability index at weak layer [−] | ssi_pwl | Profile | Discarded: correlation |
| Structural stability index at surface weak layer [−] | ssi_pwl_100 | Profile | Discarded: correlation |
| Sk38 skier stability index at weak layer [−] | sk38_pwl | Profile | Discarded: RFE |
| Sk38 skier stability index at surface weak layer [−] | sk38_pwl_100 | Profile | Discarded: correlation |
| Natural stability index at weak layer [−] | sn38_pwl | Profile | Discarded: correlation |
| Natural stability index at surface weak layer [−] | sn38_pwl_100 | Profile | Selected |
| Critical cut length at weak layer [m] | ccl_pwl | Profile | Discarded: correlation |
| Critical cut length at surface weak layer [m] | ccl_pwl_100 | Profile | Selected |
| Min. critical cut length at a deeper layer of the penentration depth [m] | min_ccl_pen | Profile | Selected |
| Skier penetration depth [cm] | pen_depth | Profile | Selected |