# Peer review of "Data-driven automated predictions of the avalanche danger level for dry-snow conditions in Switzerland"

_Natural Hazards and Earth System Sciences, 2021_

## Author Comment (AC1)

**Reply to Referee #1**

Overall, the paper is very interesting and tackles a relevant problem for the snow and avalanche community. The main methodology remains relatively simple and was already applied to different avalanche hazard data but the authors provide a deep analysis of their results to understand their algorithm behavior. In particular, they try to overcome the difficulty that their target variable (the forecasted avalanche danger) is an imperfect ground truth of the avalanche danger. The text is well written and easy to follow. The figures are of high quality. The paper is quite long but a reduction would be at the cost of completeness. My comments mainly concern minor clarifications of the methodology or some statements/findings should be qualified. I have only two major comments that should be adressed before publication.

We thank Pascal Hagenmuller for the positive review of our manuscript and the constructive comments. We will revise the paper following the suggestions. Please find below our replies (in blue).

**Major comment 1:**
In the paper, the algorithm was trained on the winter seasons 1997-1998 to 2017-2018 and evaluated on the latest two winters 2018-2019 and 2019-2020 (line 215-221). The paper findings are thus only based on these two particular years that may exhibit specific avalanche situations. I do not understand why the authors have not repeated their evaluation by extracting any two successive years in their data set and using the rest of the data for training the random forest. Therefore, I am not completely convinced that some of the presented results (some of them based on tiny differences on the evaluation scores) are perfectly robust given the high inter-annual variability of snow conditions.

We would like to thank the reviewer for this comment. The optimization process of the random forest model has been done using a 5-fold cross-validation method (please see Sections 3.4 and 4). To this end, the training data were divided into 5 subsets, in this case containing several winter seasons with an approximate size of 20 % of the training data. For each set of hyperparameters in the random grid search and the grid search, each model was tested 5 times, such that each time, one of the 5 subsets was used as a test set and the other 4 were part of the training set. The F1-score estimate was averaged over these 5 trials for each hyperparameter vector. For instance, Figure 5b shows the box plot of the F1-macro, using the 5-fold cross-validation method, with the variation of the number of features.

The two recent winter seasons were used for a final evaluation of the model's performance. These test sets, for either the $D_{tidy}$ or $D_{forecast}$, contain enough data samples for each danger level (Figure 3d).

In addition, following this suggestion, we will provide a plot of the accuracy and F1-macro of a random forest model (choosing the optimized hyperparameters and selected final features) evaluated over 10 folds with an approximate data size of 10 % (see Figure 1 below). Each test fold contains two successive winter seasons, and the remaining data are the training set. As the amount of data in the first winter seasons is considerably smaller, these first folds contain more than two winter seasons.

[Figure]

**Major comment 2:**

The input meteorological and snow data is not forecasted but derived from measurements at AWS. This is somehow expressed in section 2.1 but it appears clearly to me only when it is discussed at the end of the paper (line 536-540): the predicted avalanche danger is a nowcast and not a forecast. I think this should more clearly stated in the abstract and in the methodology as the reader can easily be mixed by « the prediction of the nowcast of the forecast ». Besides, the authors mention in the abstract (line 18-19) that a prototype was used during one winter by the Swiss avalanche warning service. However, there is no more mention of this in the paper (except the same statement in the conclusion). This is not the main scope of the paper but it is legitimate to ask how the nowcast was used/accepted by the warning service.

*We will clarify in the Abstract and Section 2.1 that the SNOWPACK simulations, and hence the model predictions, rely on measurements from AWS, and are therefore a nowcast prediction.*
*Moreover, we intend to include a short paragraph with an outlook on the setup used during operational testing of the model during the winter 2021-2022.*

**Minor comments:**

L.3-4 « based on their experience ». Not only. I guess the forecasters also follow some general guidelines as for instance, picking the right level in the EAWS bavarian matrix.

*Yes, forecasters do follow EAWS guidelines and definitions, as for instance the European Avalanche Danger Scale. However, for the final assignment of a danger level a forecaster will strongly rely on his or her experience.*

L.13 « the accuracy ». This term should be defined in the abstract or replaced by plain text, e.g. « the danger level was correctly predicted in the 72% of all cases ». Besides, the danger scale data is highly unbalanced, therefore accuracy might not be the best indicator of the algorithm performance (as explained and shown later in the paper). For instance, I can reach an accuracy of 60% by predicting always predicting 3 in Belledonne (France).

*We will reconsider using the term in the Abstract.*

L.14 « better than previously developed methods ». Remove. I think this is a bit slippery to compare to previous methods as the data, the evaluation strategy, etc. may be different.

We agree and will remove this part of the sentence.

L.16-17 « the accuracy of the current experienced-based Swiss avalanche forecasts ». I would say « agreement » instead of accuracy as we cannot certainly consider the local nowcast as a perfect ground truth too.

We agree. In this context, we will exchange the term 'accuracy' with 'agreement between forecast and nowcast assessments'.

L.23 « predicting stability in time and space ». Generally, the avalanche size is supposed to be also a characteristic of the avalanche danger.

That's correct if we talk about the definition of the danger levels. The sentence simply refers to a description of avalanche forecasting that was coined by McClung (2000).

L.28 « expert judgement » and general guidelines.

The final decision is in fact an expert judgement – in contrast to, for instance, a decision by an algorithm. Of course, the expert will consider all kind of rules, guidelines etc. for the decision.

L.47 « the only solution is to use avalanche detection systems ». No it is not the only solution, it is « another » solution. One may also take into account the uncertainty in the human based observation.

We will reword the sentence so that it becomes clear that we refer to obtaining complete avalanche catalogues.

L.68 « intrinsically noisy ». Could you please develop/explain this statement or give some references.

Thank you, we will explain the meaning of noise. Essentially, where there is judgment, there is noise (see Kahneman et al., 2021).

L.68 « danger level is the most relevant component for communicating the avalanche hazard ». Replace by « an important component ». Indeed, depending on the target public (e.g. mountain

guides), the information pyramid of the avalanche bulletin might be different (e.g. avalanche problems on top).

Please note the information pyramid used in public avalanche forecasts is the same regardless of the target audience (as shown in the EAWS recommendations; EAWS, 2021). Of course, the other elements according to the information pyramid are relevant as well, but less so overall.

L.69 « dry-snow conditions ». It might be not clear to every reader how you define dry-snow conditions. Here, I expected that you set a threshold on liquid water content. That is not the case. As far as I have understood there is always an avalanche danger level for dry snow conditions in the avalanche bulletin but sometimes there is also a wet avalanche danger scale when it is higher than the dry one. Is that correct? Please explain it somewhere in the introduction.

We will clarify the meaning. Essentially, dry-snow conditions mean that dry-snow slab avalanches are the most prominent danger. When other avalanche types become prevalent, those are specifically addressed and communicated, and when wet-snow or glide-snow avalanches dominate, the danger level refers to these avalanche types only.

L.98 and elsewhere « 1700 CET » check with the editor how you should write time in this journal. « 17:00 CET »?

We will refer to local time in the revised manuscript.

Figure 2. It appears that there can be more than one station per forecast region. How do you deal with that?

The meteorological and snowpack data from each station is an individual data sample to train and test the random forest model. The daily forecast of the region is used to label each data sample. If more than one station is in the same forecast region above the elevation indicated in the bulletin, they are assigned the same danger level label.

L.120 « the reliability, which is the trust ... as 0.9». I do not understand the number. Provide precise definition.

We will clarify in the revised manuscript what we mean when referring to reliability. The reliability of an individual danger level estimate is the scaling factor required to obtain the agreement rate of pairs of local nowcast estimates between several observers within the same warning region. For a more detailed definition, please refer to Techel (2020, Section 4.1.2, p. 35). The reliability is thus the congruence between the assessment provided by two individuals (Jacob et al., 1987), or, in other words, the factor describing the repeatability for obtaining the same danger level assessment within the same (small) warning region (Techel, 2020, p. 34). -

L.147 and 148 « accuracy ». Replace by « agreement ».

We will reword as suggested.

L.163 « level was corrected ». You mean corrected during the morning update? Clarify.

We will clarify this in the revised manuscript: "…was corrected for the purpose of this study. The danger level was not corrected during the morning forecast update, but for the purpose of this study to obtain a data set of danger level labels which corresponded best with conditions.

L.168 « High: (0.3%) » incorrect parenthesis

Thanks for spotting this.

Section 4.1. Which hyper parameters did you optimize? Number of trees, depth of the trees? And what are their final values?

We computed a random grid search and a grid search with a variable number of trees, features to consider at every split, depth of the tree, the minimum number of samples required to split a node, the minimum number of samples required at each leaf node and the maximum number of samples for each tree.

The specific hyperparameters of RF1 are:

- Number of trees =1000
- Maximum depth = 40
- Maximum features = 'log2',
- Minimum samples leaf  = 6
- Minimum samples to split =12

The specific hyperparameters of RF2 are:

- Number of trees =1000
- Maximum depth = 50
- Maximum features = ' auto',
- Minimum samples leaf  = 5
- Minimum samples to split =10

We will include the final hyperparameter settings in the Appendix.

L.257. Explain with plain text how the feature importance is computed by scikit-learn.

We will include an explanation of the feature importance computation.

L.273-274. Why did you chose 30 features since you already reached the performance plateau for 20 features?

Because with 30 features the models reached the highest scores.

L.295 « This results highlights the impact of using better-balanced training detain RF#2 and less noisy labels ». I am not convinced by this statement. Indeed, you have already indirectly balanced your data set by weighting the different classes by 1 / frequency.

We have not balanced the training dataset. We tested some balancing techniques using oversampling, undersampling, SMOTE methods, but we did not achieve an improvement of the performance. The weights are used to penalize misclassification for each class in a different way.

L. 308-314. I am wondering if the observed bias is not linked to how you weight the different classes. Do you use the same weight for both D and D_tidy even they do not contain the same frequency of danger level? Please clarify how it is done.

We used the same strategy for each data set, but independently. This means that class weights are obtained for each separate training set. Indeed, using wrong proportions could lead to biases if class separability also changes drastically depending on the learning set. Therefore, the model trained with $D_{tidy}$ has different weights than the model trained with $D_{forecast}$.

L.317 « The performance of both models improved when tested against the best possible test data ». Misleading statement (for RF2) to be changed. Indeed, you explain correctly that the RF perform at best on the set of data they were partially trained on, no link with data quality for RF2.

We agree and will modify this statement.

Section 5.3. Reading this section raised a question on the methodology. The training is done on all station together (any station.day adds a line in the data set) or is there a RF per station ? Clarify in the methods and maybe discuss these two approaches.

We will clarify when revising the manuscript. In fact, the models were trained with all the stations together. The amount of data per station varies widely as not all the stations from the IMIS network were installed at the same time or were operative in the same period of time. In addition, lower elevation stations were more often filtered due to the elevation filter used.

L.404-406. The impact of a slight distribution difference of the danger level on the overall accuracy might be quantified and I doubt that it is the reason for the geographical differences.

We have quantified the differences of the danger level forecast distribution and a discussion of the possible explanation of the geographical differences in Section 6.4.

Figure 10. Recall on the figure or in the legend the « sense » of Delta. E.g. Delta_ elevation = station elevation - bulletin elevation limit.

We will modify the Figure as suggested.

Table 3. Add the distribution of increasing, equal and decreasing danger level for each level.

We will add this information in the revised manuscript.

L420-430. Add the unit « m » when giving numbers for Delta_elevation.

We change as suggested.

L.455; « intrisically noisier ». Again give justification when you state that earlier in the text.

We will provide an explanation in the Introduction section.

L.475 « RF2 performs better on D_tidy ». Not the point here and not a justification of what is stated just before. RF2 performs better on D_tidy compared to RF1 because it is trained on D_tidy (the test subset).

Yes, we agree and will change accordingly.

L.485 « cost sensitive learning ». I am wondering whether this is somehow not equivalent to duplicating the minority classes and the following statement « reflecting the positive impact of balancing the training ratio » seems over-stated (no proof).

Cost-sensitive learning means to apply a heavier penalty on misclassifying the minority class. For this, we set the class weight as an inverse of the class frequency in the training dataset, focusing on the minority classes. This is a different technique than duplicating minority classes in the training set. Duplicating instances could be a viable technique for specific classifiers and models. For random forests specifically, duplication would have the effect of balancing the probabilities when uniformly sampling the training bags for each tree. This is equivalent to penalizing classes in the cost function, with weights proportional to the frequency of each class. Rather than duplicating, one could also sample data points according to the inverse of observed frequencies, so that no exact duplicates are present. In our analyses, this led to no benefit, and observing more data points and possibly using a large ensemble of trees is always more beneficial.

L.527. « phenomenon » . Avalanche danger is not a phenomenon.

We will reword as suggested.

Section 6.5. Clarify if the described studies apply also only to dry snow conditions.

All three studies mentioned focused as well on dry-snow conditions. We will clarify this.

Conclusion. Mention the fact that for the moment it is only a nowcast tool.

We will emphasize that we present results for the nowcast mode of the model. We have already tested the model in forecast mode, and it performed equally well (as presented by Perez et al., 2021). We will include a short outlook paragraph on the potential for running the model in forecast mode in the Discussion section.

**References:**

EAWS, 2021; https://www.avalanches.org/downloads/#informationpyramid

SLF, 2021; https://www.slf.ch/en/avalanche-bulletin-and-snow-situation/about-the-avalanche-bulletin/interpretation-guide.html

Hutter, V., Techel, F., and Purves, R. S.: How is avalanche danger described in textual descriptions in avalanche forecasts in Switzerland? Consistency between forecasters and avalanche danger, Nat. Hazards Earth Syst. Sci., 21, 3879–3897, https://doi.org/10.5194/nhess-21-3879-2021, 2021.

Kahneman, D., Sibony, O., and Sunstein, C. R.: Noise - A flaw in human judgment, Hachette Book Group, New York, U.S.A., 454 pp., 2021.

Pérez-Guillén, C., Techel, F., Hendrick, M., Volpi, M., van Herwijnen, A., and Schweizer, J.: Operational test of automatic danger level predictions in Switzerland. Colorado Snow and Avalanche Workshop, 14-15 October, 2021.

---

## Author Comment (AC2)

**Reply to Referee #2**

**General comments**

The paper presents the development of a machine-learning model capable of assessing the avalanche danger level based on input data from automatic weather stations and a snowpack model in the Swiss Alps. The models are trained using a large data set of forecasted danger levels and a filtered subset of "re-assessed" danger levels from local nowcasts.

Compared to previous studies the presented paper uses a much larger and well-refined data set. The trained machine-learning models achieve performances comparable to human forecasters throughout the region of the Swiss Alps. Previous studies did either have either poorer performance or were more limited in their spatial extend.

The topic is of scientific interest and value for avalanche researchers, forecasting services and stakeholders. The topic is within the scope of NHESS. The authors present their study in a clear manner. The manuscript is well written and structured. The abstract provides a good summary of the goals, methods and conclusions of the presented study.

Tables and figures are of high quality and readability contributing to the good overall impression of the paper. The language is precise and understandable. The paper is long. However, it combines the field of avalanche forecasting and machine- learning using the Random Forest algorithm and needs to (and does) explain both concepts to the reader potentially being unfamiliar with one or both of them. I therefore only have minor suggestion on how to shorten it - see specific comments.

It is not clear from this paper how you apply or intend to apply the model in a forecasting setting since it is trained and run on input data measured and modeled at an automatic weather station. I also miss a discussion on the how to apply the models in an operational setting and the expected benefits in supporting the human avalanche forecaster - see specific comments.

We thank Karsten Müller for his positive evaluation of our manuscript and the constructive comments. We will revise the paper following the suggestions. Please find below our replies (in blue).

**Specific comments**

l-171 Your models are trained on station data. That means they require a measurement and a subsequent SNOWPACK model output to be applied. Thus, RF#1 and RF#2 as described in this paper only provide a hindcast or nowcast.

Yes, we agree. We will clarify in the new version of the manuscript that the models presently provide a nowcast.

In order to be used operational your models need be run with input data from weather prediction models and the corresponding output from SNOWPACK at the location of IMIS stations. As far as I can see this is not addressed in your paper. Please add or reference information on how this is or could be done. I expect that the transition from the spatial resolution of the weather

model to the station site (especially in mountainous terrain) poses some scaling issues which might have an effect on performance/accuracy. This should be addressed in the discussion e.g. in connection to section 6.3.

We will emphasize that we present results for the nowcast mode of the model. We have already tested the model in forecast mode and it performed equally well (as presented by Perez et al., 2021). We will include a short outlook paragraph on the potential for running the model in forecast mode in the Discussion section.

l-207 Why do you only filter by elevation and not by aspect? I assume you do not filter by aspect because most (all used?) IMIS stations are on a flat field and thus cannot be assigned an aspect. Please add a short explanation.

The data sets for training and testing the models were computed with SNOWPACK simulations for the "flat" field. We have also tested the models with input data from SNOWPACK simulations computed for virtual slopes with different aspects (N, E, S, W). The results obtained for the different aspect are, however, outside of the scope of this paper. We will clarify in the revised manuscript that SNOWPACK simulations were computed for level terrain.

l-216 It seems legitimate to use the most recent winter seasons as test data. However, it should be ensured and stated that these do not exhibit any special avalanche conditions not or barely seen during previous winters - have you considered/tested a random draw from all data with an equal amount from each month as an alternative? If yes, what was the effect on model accuracy.

We discarded random train/test to avoid correlated data from closer stations and days to be in the training and test sets at the same time. The random forest model was optimized by using a 5-fold cross-validation method (please see Sections 3.4 and 4). To this end, the training data were divided into 5 subsets, in this case containing several winter seasons with an approximate size of 20 % of the training data. The two recent winter seasons were used for a final evaluation of the model's performance. This test sets, for either the $D_{tidy}$ or $D_{forecast}$, contain enough data samples for each danger level (Figure 3d).

Please refer to the reply to Referee #1 where we address this issue in more detail, please also see the Figure we provide there.

l-275 "Note that this last step..." - what do you mean by this sentence? It is not clear to me to which "last step" you refer and what the effect on model performance is. Could you clarify?

We refer to the step of feature selection by Recursive Feature Elimination (RFE) as described in that paragraph. In the sentence we provide an explanation why we used RFE rather than relying on internal feature ranking. We clarify this issue in the revised manuscript.

l-355 While the section "Exemplary case studies" is useful for the reader in order to get an overview over potential model outcomes in relation to published avalanche forecasts, it is not

necessary for the understanding of the paper. Considering that the paper is already very long, I suggest to move this section and Fig.8 to the Appendix or provide it as supplementary material.

Thanks, we will consider this suggestion. For the time being, we think it makes sense to keep this section in the main part of the manuscript. These cases are illustrative examples of the model output and show the overall performance in different situations. In addition, it helps to interpret the videos provided as supplementary material.

l-328 What is the "daily averaged accuracy"? Is it the average of the predictions from RF#1 and RF#2 or is it the average of the results from all stations within a forecasting region with regard to Dforecast for that region?

We will remove "averaged" as it was computed the daily accuracy per model.

l-405 The last two sentences in this section should be revised. I understand it such that performance was lower because the danger levels (1 and 3) - that have highest prediction performance - are less common in these regions. However, I had to read it several times to understand what you mean.

We will clarify these statements in the revised manuscript.

It would also be interesting to know if you could identify common traits for stations/sites that had a high accuracy (e.g. >0.8): specific elevations, typical snow or weather conditions?

Thank you for this interesting suggestion. However, the paper is already long and we prefer to provide more insight into site-specific performance in a future publication.

l-540 see comment for l-171

We will include more details about the setup in forecast mode.

l-573 Your features include several stability indices and information on weak layers. Does that mean the provided stability information from SNOWPACK is not good enough to detect/predict persistent weak layers or the stability related to them?

As mentioned, it is presently not fully clear where the regional differences stem from. In fact, our model does include stability information, but a concurrent study has recently shown that for stability prediction actually better predictor variables exist than, for instance, the skier stability index provided by SNOWPACK (Mayer et al., 2021).

l-591 Could you discuss the intended operational application of the models and their main benefits to the human forecaster in more depth. I could imagine that the models would be useful in deciding when to increase or decrease the danger level and to assess the spatial or temporal extend of a given danger level.

As mentioned above, we will include a short outlook paragraph on the setup used during operational testing of the model during the winter 2021-2022

l-602 It would also be interesting to know in the discussion what your expectations on model performance are. I would argue that your results are as best as it can get. You state that a human forecaster has an average accuracy of 76%. You use the assessment by the human forecaster as your labels. Thus, the model inherits human mistakes and biases. For RF#2 these biases are somewhat corrected for or at least replaced by biases or mistakes in human assessed nowcasts.

We agree. As we measured the model's performance using a noisy target variable (forecast danger level, "tidy" danger level), we cannot expect obtaining a model accuracy that is higher than the (unknown) errors in the forecast. Even a perfect model would show (seemingly) mediocre performance if compared with a ground truth which is very noisy. We explicitly address this issue in Section 5.2 and also illustrate it with the exemplary case studies.

l-603 It is not clear from your paper that your model "predicts" avalanche danger. I read it that your model can be used to validate or quality control a published forecast once data has been measured at an IMIS station.

We agree. However, we have already used the model in forecast mode with good success. We will shortly describe this in a new outlook paragraph as mentioned above.

l-610 see comment for l-573

Please, see reply to your comment above.

**Technical comments**

l-141 "...which jointly account for more than 75% of the cases." Change to "which jointly account for 77% of the cases."

We will change as suggested.

Fig.3 ideally the y-axis of the DL proportion [%] plot for Dforecast would have the same maximum value - currently these are 50% and 40%.

We will modify the Figure as suggested.

l-311 "...the two models...", missing "s" l-318 remove one "particularly"

l-422 spelling "Eq. 1"

Thank you for spotting these typos.

l-463 Split this sentence in two.

We will reword the sentence.

l-474 "...only the 10%..." - remove "the"

We will change as suggested.

l-581 Change to "..., predicting high probabilities for both danger levels."

We will change as suggested.

l-587 remove one "the" and the end of the line

We will change as suggested.

**References**

Mayer, S., van Herwijnen, A., and Schweizer, J.: A random forest model to assess snow instability from simulated snow stratigraphy, EGU General Assembly 2021, online, EGU21-12259, https://doi.org/10.5194/egusphere-egu21-12259, 2021.

Pérez-Guillén, C., Techel, F., Hendrick, M., Volpi, M., van Herwijnen, A., and Schweizer, J.: Operational test of automatic danger level predictions in Switzerland. Colorado Snow and Avalanche Workshop, 14-15 October, 2021.

---

## Author Response (AR1)

Dear Pascal Haegeli,

We would like to thank to editor and the two referees, Pascal Hagenmuller and Karsten Müller, for their careful review and constructive comments that have contributed to improve our manuscript. We have modified the paper following their suggestions. The changes are incorporated in the revised version of the manuscript. A point-by-point response to all the comments is provided below (in blue) and a final list of the major changes made. In addition, we provide a marked-up manuscript version showing the changes made. We hope that the manuscript is now suitable for publication in Natural Hazards and Earth System Sciences.

I look forward to hearing from you.

Sincerely,

Cristina Pérez Guillén

(on behalf of all authors)

**Reply to Editor's comments:**

1) Both reviewers mentioned that they initially thought that the algorithm provides a true forecast instead of just a nowcast. I recommend that in addition to the described revisions, you carefully review your entire manuscript and edit your wording to avoid this confusion for future readers. Replacing the term 'predict' with 'assess' or similar will likely help to make the scope of your current study more obvious right away. The potential use of the approach for predicting avalanche danger should only be mentioned in the proposed outlook paragraph.

We agree and have carefully reviewed the manuscript and clarified that the models provide a nowcast of the danger level. We still refer to "predictions" in the manuscript in the context of model outputs, as random forest classifiers predict a class membership, which in this case is the danger level. In addition, we have added a new section in the discussion where we describe the operational testing of the models (Section 6.6).

2) Both reviewers commented on the length of the manuscript, and one of them provided explicit suggestions for how to possibly shorten the paper (e.g., move the exemplary case studies section into the supplementary material). To make the paper more easily digestible for future readers, I recommend that you critically review your manuscript again and explore possibilities for reducing its length. Your response to the comment of reviewer #2 ("… it helps to interpret the videos provided as supplementary material …") makes me think that moving these illustrative examples (probably better word than exemplary case studies) could in fact be moved to the supplementary material. This is particularly important because the reviewer have requested important additional material (e.g., description of operational application).

We agree and now provide the illustrative case studies in Appendix C.

3) I agree with the reviewers that there are considerable established guidelines for picking avalanche danger ratings and this should be mentioned explicitly in the text and not assumed that readers know. This will give readers less familiar with avalanche forecasting practices a better and more accurate understanding of the situation.

We agree and now provide two examples (lines 30-32 of the marked-up manuscript).

4) Since the study is focused on assessing avalanche danger, the definition of avalanche forecasting by McClung (2000) does not seem overly fitting (Reviewer 1 – L 23 comment). Including a definition of avalanche danger might be more appropriate.

We agree and now provide an alternative definition of avalanche forecasting (lines 24-25).

5) I agree with the L 68 comment of Reviewer 1 that the danger level is not the most relevant component for communicating avalanche hazard. However, I also agree with you that the

information pyramid is the same for all users since it is a characteristic of the product and not its use. Nevertheless, a wording like "an important component of the avalanche bulletin that communicates the seriousness of existing hazard most concisely" would address the reviewer's concern and describe the role of the danger rating more accurately.

We agree that the avalanche danger level is only one component of a forecast and that for most advanced users it is not the most important one. However, for the general public, it remains the key component to convey the hazard in the sense of a warning. We have modified this sentence (lines 75 -76).

6) Regarding the L 273-274 comment by Reviewer 1: As highlighted by Reviewer #1, the performance does not seem to increase significantly anymore for models with more than 20 features. I understand from your answer that including 30 features resulted in the highest scores, but if it does not really make a difference, wouldn't a more parsimonious model be more desirable?

We agree that a simplified model may have some benefits in case some variables are not available. However, we do not see an advantage in reducing the number of features even though the loss in performance would be minor as the additional computational power needed is negligible.

**List of relevant changes:**

- Clarification that the models provide a nowcast of the danger level and not a forecast.

- According to suggestions, we define and clarify different terms (avalanche forecasting, guidelines, and more).

- Shorten the length of the manuscript, moving the illustrative cases studies to Appendix C.

- We include a new sub-section in the discussion, Section 6.6, which describes the operational testing of the models in nowcast and forecast mode.

- Modification of Figure 3 and Table 3.

- New table B1 in Appendix B.

**Reply to Referee #1**

We thank Pascal Hagenmuller for the positive review of our manuscript and the constructive comments. We have revised the paper following the suggestions. Please find below our replies (in blue). The changes are referred to the line numbers in the marked-up manuscript version.

**Major comment 1:**

In the paper, the algorithm was trained on the winter seasons 1997-1998 to 2017-2018 and evaluated on the latest two winters 2018-2019 and 2019-2020 (line 215-221). The paper findings are thus only based on these two particular years that may exhibit specific avalanche situations. I do not understand why the authors have not repeated their evaluation by extracting any two successive years in their data set and using the rest of the data for training the random forest. Therefore, I am not completely convinced that some of the presented results (some of them based on tiny differences on the evaluation scores) are perfectly robust given the high inter-annual variability of snow conditions.

We would like to thank the reviewer for this comment. The optimization process of the random forest model has been done using a 5-fold cross-validation method (please see Sections 3.4 and 4). To this end, the training data were divided into 5 subsets, in this case containing several winter seasons with an approximate size of 20 % of the training data. For each set of hyperparameters in the random grid search and the grid search, each model was tested 5 times, such that each time, one of the 5 subsets was used as a test set and the other 4 were part of the training set. The F1-score estimate was averaged over these 5 trials for each hyperparameter vector. For instance, Figure 5b shows the box plot of the F1-macro, using the 5-fold cross-validation method, with the variation of the number of features.

The two recent winter seasons were used for a final evaluation of the model's performance. These test sets, for either the $D_{tidy}$ or $D_{forecast}$, contain enough data samples for each danger level (Figure 3d).

In addition, following your suggestion, we provide below a plot of the accuracy and F1-macro of a random forest model (choosing the optimized hyperparameters and selected final features) evaluated over 10 folds with an approximate data size of 10 % (see Figure 1 below). Each test fold contains two successive winter seasons, and the remaining data are the training set. As the amount of data in the first winter seasons is considerably smaller, these first folds contain more than two winter seasons.

[Figure]

**Major comment 2:**

The input meteorological and snow data is not forecasted but derived from measurements at AWS. This is somehow expressed in section 2.1 but it appears clearly to me only when it is discussed at the end of the paper (line 536-540): the predicted avalanche danger is a nowcast and not a forecast. I think this should more clearly stated in the abstract and in the methodology as the reader can easily be mixed by « the prediction of the nowcast of the forecast ». Besides, the authors mention in the abstract (line 18-19) that a prototype was used during one winter by the Swiss avalanche warning service. However, there is no more mention of this in the paper (except the same statement in the conclusion). This is not the main scope of the paper but it is legitimate to ask how the nowcast was used/accepted by the warning service.

We now clarify in the revised manuscript that the SNOWPACK simulations, and hence the model predictions, rely on measurements from AWS, and are therefore a nowcast prediction. We now include a new section in the discussion, Section 6.6, where we provide an outlook on the operational testing of the models for avalanche forecasting.

**Minor comments:**

L.3-4 « based on their experience ». Not only. I guess the forecasters also follow some general guidelines as for instance, picking the right level in the EAWS bavarian matrix.

Yes, forecasters do follow EAWS guidelines and definitions, as for instance the European Avalanche Danger Scale. However, for the final assignment of a danger level a forecaster will strongly rely on his or her experience. We have clarified this point in the introduction (lines 30-32).

L.13 « the accuracy ». This term should be defined in the abstract or replaced by plain text, e.g. « the danger level was correctly predicted in the 72% of all cases ». Besides, the danger scale data is highly unbalanced, therefore accuracy might not be the best indicator of the algorithm performance (as explained and shown later in the paper). For instance, I can reach an accuracy of 60% by predicting always predicting 3 in Belledonne (France).

We now define this term in the Abstract (line 13).

L.14 « better than previously developed methods ». Remove. I think this is a bit slippery to compare to previous methods as the data, the evaluation strategy, etc. may be different.

We agree and removed this part of the sentence (lines 14-15).

L.16-17 « the accuracy of the current experienced-based Swiss avalanche forecasts ». I would say « agreement » instead of accuracy as we cannot certainly consider the local nowcast as a perfect ground truth too.

We agree. In this context, we exchanged the term 'accuracy' with 'agreement between forecast and nowcast assessments' (line 17; and elsewhere in the manuscript).

L.23 « predicting stability in time and space ». Generally, the avalanche size is supposed to be also a characteristic of the avalanche danger.

We now provide an alternative definition of avalanche forecasting (lines 24-25).

L.28 « expert judgement » and general guidelines.

We now clarify that avalanche forecasters follow general guidelines (lines 30-32).

L.47 « the only solution is to use avalanche detection systems ». No it is not the only solution, it is « another » solution. One may also take into account the uncertainty in the human based observation.

We reworded the sentence (line 52).

L.68 « intrinsically noisy ». Could you please develop/explain this statement or give some references.

Thank you, we now explain the meaning of noise (line 74). Essentially, where there is judgment, there is noise (see Kahneman et al., 2021).

L.68 « danger level is the most relevant component for communicating the avalanche hazard ». Replace by « an important component ». Indeed, depending on the target public (e.g. mountain guides), the information pyramid of the avalanche bulletin might be different (e.g. avalanche problems on top).

Please note the information pyramid used in public avalanche forecasts is the same regardless of the target audience (as shown in the EAWS recommendations; EAWS, 2021). Of course, the other elements according to the information pyramid are relevant as well, and for some users may be even more relevant, but for warning the general public, it remains the key component. We have reworded this sentence (line 75-76).

L.69 « dry-snow conditions ». It might be not clear to every reader how you define dry- snow conditions. Here, I expected that you set a threshold on liquid water content. That is not the case. As far as I have understood there is always an avalanche danger level for dry snow conditions in the avalanche bulletin but sometimes there is also a wet avalanche danger scale when it is higher than the dry one. Is that correct? Please explain it somewhere in the introduction.

We now clarify the meaning (lines 76-77). Essentially, dry-snow conditions mean that dry-snow slab avalanches are the most prominent danger. When other avalanche types become prevalent, those are specifically addressed and communicated, and when wet-snow or glide-snow avalanches dominate, the danger level refers to these avalanche types only.

L.98 and elsewhere « 1700 CET » check with the editor how you should write time in this journal. « 17:00 CET »?

We now refer to local time in the revised manuscript.

Figure 2. It appears that there can be more than one station per forecast region. How do you deal with that?

The meteorological and snowpack data from each station is an individual data sample to train and test the random forest model. The daily forecast of the region is used to label each data sample. If more than one station is in the same forecast region above the elevation indicated in the bulletin, they are assigned the same danger level label.

L.120 « the reliability, which is the trust ... as 0.9». I do not understand the number. Provide precise definition.

We now clarify in the revised manuscript what we mean when referring to reliability (lines 130-134). The reliability of an individual danger level estimate is the scaling factor required to obtain the agreement rate of pairs of local nowcast estimates between several observers within the same warning region. For a more detailed definition, please refer to Techel (2020, Section 4.1.2, p. 35). The reliability is thus the congruence between the assessment provided by two individuals (Jacob et al., 1987), or, in other words, the factor describing the repeatability for obtaining the same danger level assessment within the same (small) warning region (Techel, 2020, p. 34).

L.147 and 148 « accuracy ». Replace by « agreement ».

We reworded as suggested (line 163).

L.163 « level was corrected ». You mean corrected during the morning update? Clarify.

We now clarify this in the revised manuscript (line 178). The danger level was not corrected during the morning forecast update, but for the purpose of this study to obtain a data set of danger level labels which corresponded best with the actual conditions.

L.168 « High: (0.3%) » incorrect parenthesis

Thanks for spotting this.

Section 4.1. Which hyper parameters did you optimize? Number of trees, depth of the trees? And what are their final values?

We computed a random grid search and a grid search with a variable number of trees, features to consider at every split, depth of the tree, the minimum number of samples required to split a node, the minimum number of samples required at each leaf node and the maximum number of samples for each tree. We have included the final hyperparameter settings in Table B1 of Appendix B.

L.257. Explain with plain text how the feature importance is computed by scikit-learn.

We now include an explanation of the feature importance computation (lines 278-279).

L.273-274. Why did you choose 30 features since you already reached the performance plateau for 20 features?

With 30 features the models reached the highest scores.

L.295 « These results highlights the impact of using better-balanced training detain RF#2 and less noisy labels ». I am not convinced by this statement. Indeed, you have already indirectly balanced your data set by weighting the different classes by 1 / frequency.

We have not balanced the training dataset. We tested some balancing techniques using oversampling, undersampling, SMOTE methods, but we did not achieve an improvement of the performance. The weights are used to penalize misclassification for each class in a different way.

L. 308-314. I am wondering if the observed bias is not linked to how you weight the different classes. Do you use the same weight for both D and D_tidy even they do not contain the same frequency of danger level? Please clarify how it is done.

We used the same strategy for each data set, but independently. This means that class weights are obtained for each separate training set. Indeed, using wrong proportions could lead to biases if class separability also changes drastically depending on the learning set. Therefore, the model trained with $D_{\text{tidy}}$ has different weights than the model trained with $D_{\text{forecast}}$.

L.317 « The performance of both models improved when tested against the best possible test data ». Misleading statement (for RF2) to be changed. Indeed, you explain correctly that the RF

perform at best on the set of data they were partially trained on, no link with data quality for RF2.

We agree and modified this statement (lines 338-339).

Section 5.3. Reading this section raised a question on the methodology. The training is done on all station together (any station.day adds a line in the data set) or is there a RF per station ? Clarify in the methods and maybe discuss these two approaches.

We now clarify this aspect in the manuscript (lines 221-222). In fact, the models were trained with all the stations together. The amount of data per station varies widely as not all the stations from the IMIS network were installed at the same time or were operative in the same period of time. In addition, lower elevation stations were more often filtered due to the elevation filter used.

L.404-406. The impact of a slight distribution difference of the danger level on the overall accuracy might be quantified and I doubt that it is the reason for the geographical differences.

We have quantified the differences of the danger level forecast distribution and a discussion of the possible explanation of the geographical differences in Section 6.4.

Figure 10. Recall on the figure or in the legend the « sense » of Delta. E.g. Delta_ elevation = station elevation - bulletin elevation limit.

We modified the legend of Figure 9 as suggested.

Table 3. Add the distribution of increasing, equal and decreasing danger level for each level.

We added this information in Table 3.

L420-430. Add the unit « m » when giving numbers for Delta_elevation.

Changed as suggested.

L.455; « intrisically noisier ». Again give justification when you state that earlier in the text.

We have provided an explanation in the Introduction section (line 74).

L.475 « RF2 performs better on D_tidy ». Not the point here and not a justification of what is stated just before. RF2 performs better on D_tidy compared to RF1 because it is trained on D_tidy (the test subset).

Yes, we agree and changed the statement accordingly (line 506).

L.485 « cost sensitive learning ». I am wondering whether this is somehow not equivalent to duplicating the minority classes and the following statement « reflecting the positive impact of balancing the training ratio » seems over-stated (no proof).

Cost-sensitive learning means to apply a heavier penalty on misclassifying the minority class. For this, we set the class weight as an inverse of the class frequency in the training data set, focusing on the minority classes. This is a different technique than duplicating minority classes in the training set. Duplicating instances could be a viable technique for specific classifiers and models. For random forests specifically, duplication would have the effect of balancing the probabilities when uniformly sampling the training bags for each tree. This is equivalent to penalizing classes in the cost function, with weights proportional to the frequency of each class. Rather than duplicating, one could also sample data points according to the inverse of observed frequencies, so that no exact duplicates are present. In our analyses, this led to no benefit, and observing more data points and possibly using a large ensemble of trees is always more beneficial.

L.527. « phenomenon ». Avalanche danger is not a phenomenon.

We reworded as suggested (lines 556-557).

Section 6.5. Clarify if the described studies apply also only to dry snow conditions.

All three studies mentioned focused as well on dry-snow conditions. We now clarify this (line 607).

Conclusion. Mention the fact that for the moment it is only a nowcast tool.

We have emphasized in the conclusions (line 649) and the rest of the paper that we present results for the nowcast mode of the model. We have already tested the model in forecast mode, and it performed equally well (as presented by Perez et al., 2021). We now include a short outlook on the operational testing of the models in forecast mode in Section 6.6.

**References:**

EAWS, 2021; https://www.avalanches.org/downloads/#informationpyramid
SLF, 2021; https://www.slf.ch/en/avalanche-bulletin-and-snow-situation/about-the-avalanche-bulletin/interpretation-guide.html

Hutter, V., Techel, F., and Purves, R. S.: How is avalanche danger described in textual descriptions in avalanche forecasts in Switzerland? Consistency between forecasters and avalanche danger, Nat. Hazards Earth Syst. Sci., 21, 3879–3897, https://doi.org/10.5194/nhess-21-3879-2021, 2021.

Kahneman, D., Sibony, O., and Sunstein, C. R.: Noise - A flaw in human judgment, Hachette Book Group, New York, U.S.A., 454 pp., 2021.

Pérez-Guillén, C., Techel, F., Hendrick, M., Volpi, M., van Herwijnen, A., and Schweizer, J.: Operational test of automatic danger level predictions in Switzerland. Colorado Snow and Avalanche Workshop, 14-15 October, 2021.

**Reply to Referee #2**

We thank Karsten Müller for his positive evaluation of our manuscript and the constructive comments. We have revised the paper following the suggestions. Please find below our replies (in blue).

**Specific comments**

l-171 Your models are trained on station data. That means they require a measurement and a subsequent SNOWPACK model output to be applied. Thus, RF#1 and RF#2 as described in this paper only provide a hindcast or nowcast.

Yes, we agree. We now clarify in the revised version of the manuscript that the models presently provide a nowcast.

In order to be used operational your models need be run with input data from weather prediction models and the corresponding output from SNOWPACK at the location of IMIS stations. As far as I can see this is not addressed in your paper. Please add or reference information on how this is or could be done. I expect that the transition from the spatial resolution of the weather model to the station site (especially in mountainous terrain) poses some scaling issues which might have an effect on performance/accuracy. This should be addressed in the discussion e.g. in connection to section 6.3.

We now emphasize that we present results for the nowcast mode of the model. We have already tested the model in forecast mode and it performed equally well (as presented by Perez et al., 2021). We now include a short outlook paragraph on the potential for running the model in forecast mode in a new section of the Discussion (Section 6.6).

l-207 Why do you only filter by elevation and not by aspect? I assume you do not filter by aspect because most (all used?) IMIS stations are on a flat field and thus cannot be assigned an aspect. Please add a short explanation.

The data sets for training and testing the models were computed with SNOWPACK simulations for the "flat" field. We have also tested the models with input data from SNOWPACK simulations computed for virtual slopes with different aspects (Section 6.6). We now clarify in the revised manuscript that SNOWPACK simulations were computed for level terrain (lines 103; lines 227-228).

l-216 It seems legitimate to use the most recent winter seasons as test data. However, it should be ensured and stated that these do not exhibit any special avalanche conditions not or barely seen during previous winters - have you considered/tested a random draw from all data with an equal amount from each month as an alternative? If yes, what was the effect on model accuracy.

We discarded random train/test to avoid correlated data from closer stations and days to be in the training and test sets at the same time. The random forest model was optimized by using a 5-fold cross-validation method (please see Sections 3.4 and 4). To this end, the training data were divided into 5 subsets, in this case containing several winter seasons with an approximate size of 20 % of the training data. The two recent winter seasons were used for a final evaluation of the model's performance. This test sets, for either the $D_{tidy}$ or $D_{forecast}$, contain enough data samples for each danger level (Figure 3d).

Please refer to the reply to Referee #1 where we address this issue in more detail, please also see the Figure we provide there.

l-275 "Note that this last step..." - what do you mean by this sentence? It is not clear to me to which "last step" you refer and what the effect on model performance is. Could you clarify?

We refer to the step of feature selection by Recursive Feature Elimination (RFE) as described in that paragraph. In the sentence we provide an explanation why we used RFE rather than relying on internal feature ranking.

l-355 While the section "Exemplary case studies" is useful for the reader in order to get an overview over potential model outcomes in relation to published avalanche forecasts, it is not necessary for the understanding of the paper. Considering that the paper is already very long, I suggest to move this section and Fig.8 to the Appendix or provide it as supplementary material.

We agree and now provide this section as Appendix C.

l-328 What is the "daily averaged accuracy"? Is it the average of the predictions from RF#1 and RF#2 or is it the average of the results from all stations within a forecasting region with regard to Dforecast for that region?

We removed "averaged" (lines 349 and caption of Figure 7) as the daily accuracy per model was computed.

l-405 The last two sentences in this section should be revised. I understand it such that performance was lower because the danger levels (1 and 3) - that have highest prediction performance - are less common in these regions. However, I had to read it several times to understand what you mean.

We reworded these sentences (lines 433-435).

It would also be interesting to know if you could identify common traits for stations/sites that had a high accuracy (e.g. >0.8): specific elevations, typical snow or weather conditions?

Thank you for this interesting suggestion. However, the paper is already long and we prefer to provide more insight into site-specific performance in a future publication.

l-540 see comment for l-171

We now include more details about the setup in forecast mode in the new Section 6.6.

l-573 Your features include several stability indices and information on weak layers. Does that mean the provided stability information from SNOWPACK is not good enough to detect/predict persistent weak layers or the stability related to them?

As mentioned, it is presently not fully clear where the regional differences stem from. In fact, our model does include stability information, but a concurrent study has recently shown that for stability prediction actually better predictor variables exist than, for instance, the skier stability index provided by SNOWPACK (Mayer et al., 2021).

l-591 Could you discuss the intended operational application of the models and their main benefits to the human forecaster in more depth. I could imagine that the models would be useful in deciding when to increase or decrease the danger level and to assess the spatial or temporal extend of a given danger level.

As mentioned above, we now include a short outlook paragraph on the setup used during operational testing of the model during the winter 2020-2021 in Section 6.6.

l-602 It would also be interesting to know in the discussion what your expectations on model performance are. I would argue that your results are as best as it can get. You state that a human forecaster has an average accuracy of 76%. You use the assessment by the human forecaster as your labels. Thus, the model inherits human mistakes and biases. For RF#2 these biases are somewhat corrected for or at least replaced by biases or mistakes in human assessed nowcasts.

We agree. As we measured the model's performance using a noisy target variable (forecast danger level, "tidy" danger level), we cannot expect obtaining a model accuracy that is higher than the (unknown) errors in the forecast. Even a perfect model would show (seemingly) mediocre performance if compared with a ground truth which is very noisy. We explicitly address this issue in Section 5.2 and also illustrate it with the exemplary case studies (Appendix C).

l-603 It is not clear from your paper that your model "predicts" avalanche danger. I read it that your model can be used to validate or quality control a published forecast once data has been measured at an IMIS station.

We agree. However, the model can also be run in forecast mode using NWP data. In fact, we have already tested the model in forecast mode with good success. We now have shortly describe this setup in a new outlook paragraph (Section 6.6) as mentioned above.

l-610 see comment for l-573

Please, see reply to your comment above.

**Technical comments**

l-141 "...which jointly account for more than 75% of the cases." Change to "which jointly account for 77% of the cases."

Changed as suggested (line 155).

Fig.3 ideally the y-axis of the DL proportion [%] plot for Dforecast would have the same maximum value - currently these are 50% and 40%.

We modified Figure 3 as suggested.

l-311 "...the two models...", missing "s" l-318 remove one "particularly"

l-422 spelling "Eq. 1"

Thank you for spotting these typos.

l-463 Split this sentence in two.

We split the sentence in two (lines 492-493).

l-474 "...only the 10%..." - remove "the"

Changed as suggested (line 504).

l-581 Change to "..., predicting high probabilities for both danger levels."

Changed as suggested (line 560).

l-587 remove one "the" and the end of the line

Changed as suggested (line 618).

**References**

Mayer, S., van Herwijnen, A., and Schweizer, J.: A random forest model to assess snow
instability from simulated snow stratigraphy, EGU General Assembly 2021, online,
EGU21-12259, https://doi.org/10.5194/egusphere-egu21-12259, 2021.

Pérez-Guillén, C., Techel, F., Hendrick, M., Volpi, M., van Herwijnen, A., and Schweizer, J.:
Operational test of automatic danger level predictions in Switzerland. Colorado Snow and
Avalanche Workshop, 14-15 October, 2021.

---

## Author Response (AR2)

Dear Pascal Haegeli,

We would like to thank you for your detailed review and constructive comments. We greatly appreciate your help in improving our manuscript. We have modified the paper following your suggestions. The changes are incorporated in the revised version of the manuscript. A point-by-point response to all the comments is provided below (in blue). In addition, we provide a marked-up manuscript version showing the changes made. We hope that the manuscript is now suitable for publication in Natural Hazards and Earth System Sciences.

I look forward to hearing from you.

Sincerely,

Cristina Pérez Guillén

(on behalf of all authors)

**Reply to Editor's comments:**

1. Overall, the writing is quite complicated with lots of long and convoluted sentences. I think that simplifying the language would make the manuscript easier to read and more accessible to readers. While I highlighted a few grammatical errors in the manuscript, I will leave the copy-editing to the editorial team at Copernicus. However, I recommend having a native English speaker proofread the manuscript before submission.

We agree and tried to simplify some parts of the manuscript. We changed the text following most of your suggestions.

2. I have some suggestions about how to better describe the source and nature of the inherent noise in avalanche danger rating datasets. The attached PDF for details. I believe that a more detailed discussion of this challenge in the introduction would be useful and eliminate the need to explain this several times throughout the manuscript.

Thanks for your suggestions. We added this description in the Introduction (Lines 70-73) and moved a paragraph of the Discussion to Section 3.1.2, where we introduce the compilation of the "tidy" data set (Lines 160-165).

3. I think it would be useful to explain the reliability of the danger rating assessments that was derived by Techel (2020) in simpler terms (L-130). The current description is hard to understand for readers not directly familiar with Frank's work.

We agree and now provide a simplified explanation (Lines 131-132).

4. One of the reviewers raised some questions about the splitting of the data set and the use of the last two winter seasons (2018/19 and 2019/20) for the validation of the model. While you reiterate your explanation from the original version of the manuscript in your response to the reviewer's comment and provided additional evidence about how the selection of the validation winters does not make a difference, you did not actually expand your explanation in the manuscript. I suspect that many NHESS readers will have similar thoughts as the reviewer, and I feel that it would be prudent to proactively address these potential questions. Hence, I suggest that you expand Section 3.4 slightly to better explain your choices and maybe provide a brief characterization of the winters that were used for validation. I am not questioning your choices, but I think that explaining them better will prevent questions from future readers.

We agree and extended the explanation about the 5-fold cross-validation method in Sections 3.4 (Lines 245-253) and 4.2 (Lines 293-297).

5. I feel similarly about your choice to select 30 features even though the performance does not seem to substantially improve after 20 features (L-295). Explicitly explaining your reasons will prevent future readers from having the same questions.

Yes, we opted to develop a model with 30 features as it has the highest scores. A simplified model would have some benefits if some data were missing, and we will consider this when

developing future models. Unlike other machine learning methods, random forest models are practically insensitive when adding less important features.

In addition, we provide below a plot comparing the scores (accuracy and F1-macro) of two random forest models trained with 20 and 30 features. We evaluated them over each fold and the final test set, showing that with 30 features the performance is overall higher than with 20 features.

[Figure]

Figure 1. Accuracy and F1-macro scores of the evaluation of two random forest models trained with the optimized hyperparameters and two different sets of features, 20 and 30 features, and used as a validation set one of the five folds (Section 3.4) and the final test set.

6. The current structure of your discussion section is very much focused on the technical aspect of model development, and much of the text seems to mainly summarize information you presented in the result section. While you describe a few important practical insights (e.g., L-534: the model performs as well as the forecasters; L-588: Model performance might be different for different avalanche situations, etc.), they seem to get lost in the technical details. However, in my opinion, these practical insights are the most important for the future operational use of these models and their future development. Hence, I wonder whether organizing your discussion more around practical questions like a) How does the model performance compare to human forecasters? b) What are the situations when the model performs poorly? c) What are the implications of the comparison between RF #1 and RF #2 for future model development in Switzerland and potentially other areas?, … instead of the technical model development aspects (e.g., Training data size and class distribution, Quality of labels, etc.) would be more informative for the NHESS readership and the avalanche community, which both consist of researchers and practitioners. See the annotations in the attached PDF for more details.

We agree and modified the Discussion section following most of your suggestions. We believe that the discussion of technical details of model development is clearly necessary and useful in this first publication. A future study will focus on the practical application and the results observed of operational testing.

We have reduced some parts of the discussion following your advice (Sections 6.1, 6.2 and 6.4). We now include a new sub-section (Section 6.6) where we address the comparison

between RF #1 and RF #2 and the future implications of the use of the models. We do not extend our discussion about the situations when the model performs poorly as it is not the scope of this paper and we have not quantitatively evaluated model performance for different avalanche problems.

**Reply to specific comments:**

- Line 153: "ground truth data labeling". Could you not just say "for model development and validation"?

We believe that the use of "data labelling" is appropriate when developing a supervised classification machine learning model; it is common and widely used in the literature. Each set of input features should be assigned to a class label in the process of model development.

- Line 178: I am not sure what the purpose of this addition is as it makes it sound that danger levels were corrected specifically for this study, which I do not think is the case.

In fact, some of the danger levels were specifically corrected for this study when compiling the "tidy" data set.

- Line 449. Please explain why you are changing from "accuracy" to "agreement rate". To me, it seems that you are still evaluating the model by comparing it to D_forecast, which is what you did in the previous sections when you used the term "accuracy". You also use the term "accuracy" in the text that describes the content of the table. If you are doing something distinctly differently, please explain in detail and justify the use of the different term.

We agree, in this case 'accuracy' is the right term.

- Line 595: You seem to be presenting new results here, which does not seem appropriate for the discussion section.

In this part of the discussion, we do not intend to present new results, we discuss the results for the regions of Davos and St. Moritz in more detail and suggest possible reasons for the observed decrease in performance in those regions.

- Line 652. While I agree with your statement, the lower danger rating at lower elevations actually does not directly derive from the lower accuracy. This makes me wonder whether

your dataset should have taken the rule "Danger rating outside of core zone one level lower" into account. Is this something to consider for future research?

We tested models labelling the stations outside the core zone with one danger level lower but the performance was lower.